# CLUTCH: Contextualized Language model for Unlocking Text-Conditioned Hand motion modelling in the wild

**Balamurugan Thambiraja**[1,2]**, Omid Taheri**[2]**, Radek Danecek**[2]**, Giorgio Becherini**[2]**,**
**Gerard Pons-Moll**[3,4,5] **& Justus Thies**[1,2]

Technical University of Darmstadt, Germany[1]
Max-Planck Institute for Intelligent Systems, Tuebingen, Germany[2]
University of Tuebingen, Germany[3]
Tubingen AI Center, Germany[4]
Max Planck Institute for Informatics, Saarland Informatics Campus, Germany[5]
`https://balamuruganthambiraja.github.io/CLUTCH`

## Abstract

Hands play a central role in daily life, yet modeling natural hand motions remains underexplored. Existing methods that tackle text-to-hand-motion generation or hand animation captioning rely on studio-captured datasets with limited actions and contexts, making them costly to scale to "in-the-wild" settings. Further, contemporary models and their training schemes struggle to capture animation fidelity with text–motion alignment. To address this, we (1) introduce '3D Hands in the Wild' (3D-HIW), a dataset of 32K 3D hand-motion sequences and aligned text, and (2) propose CLUTCH, an LLM-based hand animation system with two critical innovations: (a) SHIFT, a novel VQ-VAE architecture to tokenize hand motion, and (b) a geometric refinement stage to finetune the LLM. To build 3D-HIW, we propose a data annotation pipeline that combines vision–language models (VLMs) and state-of-the-art 3D hand trackers, and apply it to a large corpus of egocentric action videos covering a wide range of scenarios. To fully capture motion in-the-wild, CLUTCH employs SHIFT, a part–modality decomposed VQ-VAE, which improves generalization and reconstruction fidelity. Finally, to improve animation quality, we introduce a geometric refinement stage, where CLUTCH is co-supervised with a reconstruction loss applied directly to decoded hand motion parameters. Experiments demonstrate state-of-the-art performance on text-to-motion and motion-to-text tasks, establishing the first benchmark for scalable in-the-wild hand motion modelling. Code, data and models will be released.Project page:

## 1 Introduction

Hands are at the heart of our daily experiences: With them we write, knit, play instruments, and perform countless other actions that feel effortless to us but remain challenging for generative models to reproduce naturally. Capturing this variability is not only essential for natural motion generation, but also foundational for future behavioral AI, where models must infer, predict, and generate human behavior in interactive settings such as AR/VR, robotics, and human–computer collaboration. While prior work has focused on full-body motion, gestures, and hand–object interactions (Chen et al., 2024; Jiang et al., 2024; Liu et al., 2024; Ng et al., 2024; Huang et al., 2025; Christen et al., 2024; Cha et al., 2024; Petrov et al., 2025), text-guided hand motion generation "in-the-wild" remains underexplored, with text-to-hand–object interaction methods being the most related line of work.

Hand motion models (Huang et al., 2025; Zhou et al., 2024; Cha et al., 2024; Zhang et al., 2025b) are primarily trained on high-quality 3D hand motion datasets, such as GRAB (Taheri et al., 2020), ARCTIC (Fan et al., 2023), and H2O (Kwon et al., 2021), all captured in motion capture studios. However, collecting such datasets is both time-consuming and expensive, limiting scalability to diverse scenarios and actions. As a result, current methods are restricted to a narrow set of actions

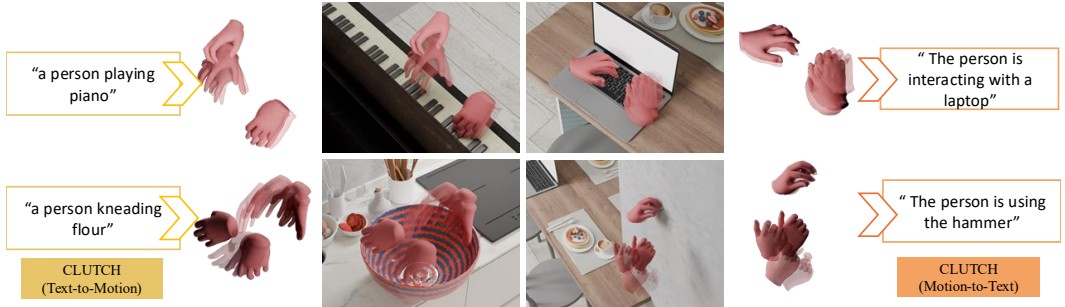

Figure 1: CLUTCH is a novel LLM-based model that enables text-conditioned synthesis (left) and captioning of in-the-wild 3D hand motions (right).

and intents, and cannot generate "in-the-wild" motions. To mitigate this data limitation, we draw inspiration from prior work (Wang et al., 2025; Sklyarova et al., 2023), which leverage VLMs/LLMs as data annotators. Specifically, we integrate a 3D hand tracker (Zhang et al., 2025a) with a VLM (Wu et al., 2024) to construct an "in-the-wild" hand motion dataset comprising 32K sequences; approximately $10\times$ larger than GRAB and ARCTIC, and $2\times$ larger than the recent Gigahands (Fu et al., 2025) dataset. We refer to this dataset as **'3D Hands in the Wild' (3D-HIW)** dataset, which includes multi-action clips like piano and food prep, underrepresented in previous work.

While VLMs demonstrate strong visual understanding, they often hallucinate spurious objects, actions, or concepts (Wu et al., 2025) when captioning. To address this, we introduce a *Parallelized Chain-of-Thought Prompting strategy*, which decomposes a complex reasoning prompt into multiple atomic prompts, each targeting a specific video aspect. The atomic responses are processed by a summarization module to generate an initial annotation, then refined into a more detailed annotation.

Compared to most existing hand motion datasets, which mostly contain single actions or interactions per sequence, in-the-wild hand movements are more natural and diverse, often involving multiple actions within the same sequence. This requires a motion model that can robustly align hand motion with language representations. Recent approaches, HOIGPT (Huang et al., 2025) and MotionGPT (Huang et al., 2025), repurpose pre-trained LLMs for motion tasks. However, we find that applying them as-is to hand animation leads to suboptimal performance due to (1) poor generalization capability of the motion tokenizer, and (2) geometric inaccuracies in the LLM-predicted motion. We address this by introducing **CLUTCH (Contextualized Language model for Unlocking Text-Conditioned Hand motion)**, a novel LLM for synthesizing and captioning in-the-wild 3D hand motions (illustrated in Fig. 1). In CLUTCH, we address the aforementioned limitations by: (1) a novel hand motion prior and (2) a new LLM finetuning stage with a geometric refinement loss.

**(1) Motion prior.** Hand motions are inherently multi-modal. Using a standard single VQ-VAE for both hands leads to poor quality of hand motion reconstruction (jitter or lack of realism). The diversity of hand motions observed "in-the-wild" exposes this issue further. To address this, we introduce **SHIFT (Structuring Hands Into Fine-grained Tokens)**. SHIFT models trajectory and pose components using separate VQ-VAE's, while disentangling left and right hands during encoding and decoding. Empirically, this formulation achieves stronger generalization and more accurate reconstructions, even under high temporal compression compared to a standard VQ-VAE. It also improves bimanual coordination and reduces jitter.

**(2) LLM finetuning.** We find that finetuning the LLM on the standard next-token prediction task with the cross-entropy (CE) loss leads to suboptimal animation fidelity. We find that token-level accuracy does not guarantee high-quality motion synthesis (as shown in (Hong et al., 2024)). An additional reconstruction loss in motion space is needed to improve the motion generation. In CLUTCH, we add a novel geometry refinement stage that decodes the sampled tokens into the hand motion parameters and applies a reconstruction loss directly to the decoded hand motion parameters. This guides the LLM toward selecting codes with stronger animation fidelity. With these, CLUTCH achieves state-of-the-art on in-the-wild hand motion synthesis and captioning, and goes beyond studio captures, by generating everyday in-the-wild motions rarely seen in mocap: *playing piano (bimanual)*, *cooking*, *writing*, *knitting*, and more. We show quantitatively, that CLUTCH outperforms recent state-of-the-art methods such as HumanMDM, MotionGPT, and T2M-GPT.

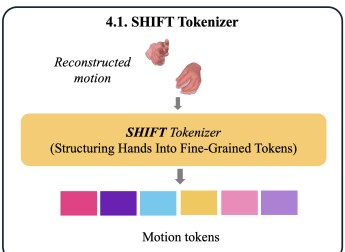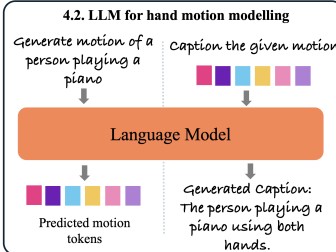

Figure 2: **Overview:** CLUTCH is an LLM for synthesizing and captioning in-the-wild 3D hand motions. To train this model, we (i) generate an in-the-wild hand motion dataset (Section 3). We (ii) tokenize the hand motion using a novel decomposed VQ-VAE tokenizer (Section 4.1). We (iii) train the LLM to model both text and motion in a unified token space (Section 4.2).

The overview of our work is presented in Figure 2. Taken together, our main contributions are:

(1) A data acquisition pipeline that combines a 3D hand tracker with a novel annotation framework driven by a vision-language model to enable scalable in-the-wild 3D hand motion data curation.

(2) Using this pipeline, we construct '3D Hands in the Wild' (3D-HIW), a large-scale dataset comprising over 32K hand motion sequences captured in diverse real-world egocentric videos.

(3) We introduce SHIFT (Structuring Hands Into Fine-grained Tokens) tokenizer, for modelling in-the-wild hand motions. SHIFT improves performance over tokenizers used in prior works.

(4) Finally, we propose CLUTCH, an LLM-based generative model for text-conditioned synthesis and captioning of in-the-wild 3D hand motions; setting a new benchmark for scalable in-the-wild hand motion modelling.

## 2 RELATED WORK

**Motion Datasets / Annotation:** Existing motion datasets provide a foundation for current human modelling methods. AMASS (Mahmood et al., 2019) unifies diverse mocap datasets into a large-scale human body motion dataset. While GRAB, ARCTIC, H2O, DexYCB (Chao et al., 2021), and OakInk (Zhan et al., 2024; Yang et al., 2022) offer detailed 3D hand–object interactions. More recently, Gigahands (Fu et al., 2025) introduced a dataset of 15K hand motion sequences with diverse actions and objects. While these datasets are of high quality, they are costly to collect, confined to controlled studio settings, and cover only narrow action ranges. In contrast, large-scale egocentric datasets such as Ego4D (Grauman et al., 2022) and EgoVid5M (Wang et al., 2024) capture diverse real-world activities but lack accurate 3D hand reconstructions and textual action descriptions. Parallel efforts in egocentric video captioning, such as LaViLa (Zhao et al., 2023), HOD (Pei et al., 2025), and EgoLM (Hong et al., 2024), leverage language models generating faithful action descriptions from input videos. LaViLa and EgoLM employ large language models (LLMs) to generate dense narrations, while HOD augments narrations (if present) with detected hand–object trajectories to produce semantically richer descriptions. To enable in the wild hand motion modelling, we construct a large-scale 3D hand motion dataset called '3D Hands in the Wild' (3D-HIW)' based on Ego4D . To this end, we introduce a two-stage annotation pipeline that first applies open-vocabulary reasoning via parallel chain-of-thought prompting, and then refines results with closed-vocabulary grounding.

**Motion Modelling:** Research in motion generation has largely focused on full-body and gesture synthesis (Guo et al., 2024; Liu et al., 2023; Zhang et al., 2023; Jiang et al., 2025; Wang et al., 2023; Shafir et al., 2023; Xie et al., 2023; Karunratanakul et al., 2023; Zhang et al., 2025c; 2023; Athanasiou et al., 2024; Chi et al., 2024; Liu et al., 2024). Parallel works have focused on hand–object interaction modelling (Christen et al., 2024; Cha et al., 2024; Ghosh et al., 2023; Zhou et al., 2022; 2024), built on MoCap datasets like GRAB (Taheri et al., 2020) or ARCTIC (Fan et al., 2023). Recent works such as (Huang et al., 2025; Jiang et al., 2024; Chen et al., 2024; Li et al., 2025a) treat motion tokens as text-like symbols, enabling pretrained LLMs to synthesize motions. While promising, these methods are limited by small-scale datasets and training objectives that emphasize token prediction accuracy rather than reconstruction fidelity. EgoLM (Hong et al., 2024) addresses this by introducing soft-linear blending regression losses during pretraining, improving text–motion alignment. However, such regression objectives conflict with cross-entropy: blending encourages smooth interpolations, whereas

Figure 3: **Data annotation pipeline:** We generate motion–text pairs from egocentric videos using a novel automated annotation framework combined with a state-of-the-art hand tracker. Text annotations are produced by first applying Parallel Chain-of-Thought prompting for open-vocabulary reasoning, followed by a closed-vocabulary refinement stage.

CE enforces sharp token choices, leading to ambiguous representations and reduced generalization. Our approach extends this line of work with a geometry-alignment stage after pretraining, where Gumbel-Softmax sampling and reconstruction losses guide the LLM toward motions that are both semantically grounded and geometrically consistent.

**VQVAE as Motion Prior:** VQ-VAE tokenizers discretize motion into language-like symbols (Jiang et al., 2024; Guo et al., 2022), but single codebooks fail to capture multi-modality. Extensions use multiple codebooks: for hand/face (Yi et al., 2023), hand/object (Huang et al., 2025), or decomposed body parts (Chen et al., 2024). (Wang et al., 2025) further explore scaling strategies to expand capacity. We build on these ideas by disentangling trajectories and hand poses into distinct codebooks, and further separate left and right hands. This yields finer control and improved generalization under temporal compression, surpassing prior single- and multi-codebook designs. A more detailed version of the related works is presented in Appendix D.

## 3   3D HANDS IN THE WILD (3D-HIW) DATASET

To enable in-the-wild hand motion modelling, we construct a large-scale 3D hand motion dataset based on in-the-wild videos from Ego4D Grauman et al. (2022) and EgoVid5M Wang et al. (2024). We propose a two-stage VLM-based text annotation and a motion reconstruction pipeline.

### 3.1   AUTOMATIC TWO-STAGE TEXT ANNOTATION PIPELINE

To generate textual descriptions from egocentric action videos, we propose an automated two-stage annotation pipeline using VLMs/LLMs. We employ VILA (Wu et al., 2024) as the VLM for its strong performance in video–language understanding and scalability for dense frame-level queries. Generating reliable annotations from egocentric videos is complicated, since the model needs to jointly reason about hand motion, user intent, and object–scene relationships. To address these challenges, we propose a two-stage pipeline, shown in Figure 3. In Stage 1 (Open-vocabulary high-level annotation), we introduce a *Parallel Chain-of-Thought* prompting strategy, which decomposes the reasoning process into several atomic prompts focused on the hand role, action–object relations, state transitions, and intent. These responses are then aggregated by a summarization LLM (Claude) to produce a coherent high-level description and reduce hallucinations. In Stage 2 (Closed-vocabulary fine-grained annotation), we refine these high-level annotations by constraining the VLM to select plausible object–action pairs from a curated vocabulary, mined from EgoVid5M and Ego4D narrations and organized into semantically meaningful clusters. This closed-vocabulary grounding improves consistency, and yields more faithful fine-grained annotations. We present the annotations generated by our method for a few sample sequences in Figure 5. Finally, we verify the generated annotations using an additional VLM pass and filter outliers with a Local Outlier Factor (LOF) filter. These refined annotations serve as supervision for the downstream training of our text-conditioned hand motion synthesis model. The prompts used for the annotation pipeline are presented in Appendix C.3.

**Stage 1: Open-vocabulary high-level annotation**

*a) Atomic prompts and responses (Parallel CoT):*
- **Hand role:** Right hand manipulates a knife; left hand stabilizes the bread.
- **Action–object relation:** Hand uses a knife to spread butter on bread.
- **State transition:** Scoop → spread → place slice → repeat on another slice.
- **Action intent:** Preparing buttered bread (food preparation).

*b) Summarization (Claude):*
*"The person spreads butter on bread slices using a knife."*

**Stage 2: Closed-vocabulary fine-grained annotation**

*a) Inputs:*
- **Stage-1 summary:** *"The person spreads butter on bread slices using a knife."*
- **Potential objects:** {bread slice, knife, butter, plate, cutting board}
- **Potential actions:** {scoop, spread, hold/stabilize, place, pick up}
- **Potential Hand roles:** {manipulator, stabilizer, both}

*b) Refinement (Claude):*
*"The right hand uses a knife to scoop butter and spread it over a bread slice while the left hand holds the slice steady."*

Figure 4: Example of the two-stage annotation pipeline for an egocentric video (Figure 5).

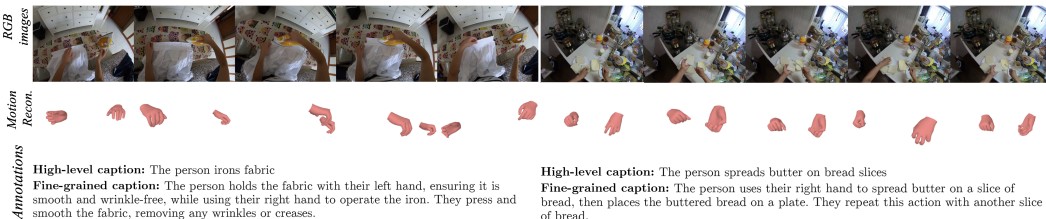

*RGB images*

*Motion Recon.*

*Annotations*

**High-level caption:** The person irons fabric
**Fine-grained caption:** The person holds the fabric with their left hand, ensuring it is smooth and wrinkle-free, while using their right hand to operate the iron. They press and smooth the fabric, removing any wrinkles or creases.

**High-level caption:** The person spreads butter on bread slices
**Fine-grained caption:** The person uses their right hand to spread butter on a slice of bread, then places the buttered bread on a plate. They repeat this action with another slice of bread.

Figure 5: Examples of the generated annotation and motion reconstruction from egocentric videos using our data annotation pipeline. For better visulaization please see the SupMat video.

## 3.2 MOTION RECONSTRUCTION

To extract 3D hand motion reconstructions from egocentric videos, we first process high-level text descriptions from the EgoVid5M dataset to identify sequences involving human presence, particularly those where humans interact with objects. We then cluster these textual descriptions into scene-level activity categories (e.g., crafting, repair) and sample sequences from each cluster to ensure diverse coverage, given that certain categories like cooking are overrepresented. Next, we run a hand keypoint tracker over the sampled videos and retain only those sequences where both hands are visible in at least 80% of the frames. We use HaWor (Zhang et al., 2025a) to reconstruct 3D hand motions from these egocentric sequences in a global coordinate frame. To reduce the noise in the reconstructed motions, we apply the Savitzky-Golay filter (Savitzky & Golay, 1964) followed by a Gaussian filter. Finally, we compute the mean of the top-3 sequence-level acceleration on both translation and rotation parameters to identify and filter out samples with abrupt, jittery transitions, indicating HaWor failures.

## 3.3 DATASET ANALYSIS

Our '3D Hands in the wild' (3D-HIW) motion dataset contains 5000 minutes of 3D hand poses and text descriptions, covering over 1355 objects and 1045 verbs. In total, 3D-HIW comprises 12M hand poses represented with MANO parameters. In Figure 6, we compare the top-200 trajectories between 3D-HIW and mocap datasets. While captured motions appear repetitive and front-facing, in-the-wild motions show greater variability in shape, end positions, and speed. t-SNE embeddings of trajectories and hand poses of top-3000 diverse samples further confirm that 3D-HIW spans a broader distribution than GRAB or Gigahands, capturing richer variability of real-world interactions. For more details, see Appendix C.1.

## 4 MOTION MODELLING

To model in-the-wild hand motions, we first tokenize the motion space into discrete tokens using a decomposed VQ-VAE.Based on this motion space, we train an LLM to model text and motion tokens in a unified latent space which allows us to do both motion synthesis from text and captioning of hand motions. **Motion parameterization:** We represent the hand motions as $\mathbf{M} = (\mathcal{H}_l, \mathcal{H}_r) \in \mathbb{R}^{D \times N}$, where $N$ represents the total number of frames in the motion, $D$ represents the motion dimension, and $l/r$ denotes the left and right hand respectively. The hand motions are parameterized using the MANO hand model (Romero et al., 2017) represented as $\mathcal{H}_j = (\tau_j, \theta_j) \in \mathbb{R}^{D/2 \times N}$, with $j \in \{l, r\}$, $\tau_j \in \mathbb{R}^{9 \times N}$ represents the trajectory of the hand motion, which contains 6D global rotation and translation. $\theta_j \in \mathbb{R}^{90 \times N}$ denotes the hand pose representing the 15 joints with 6D rotation.

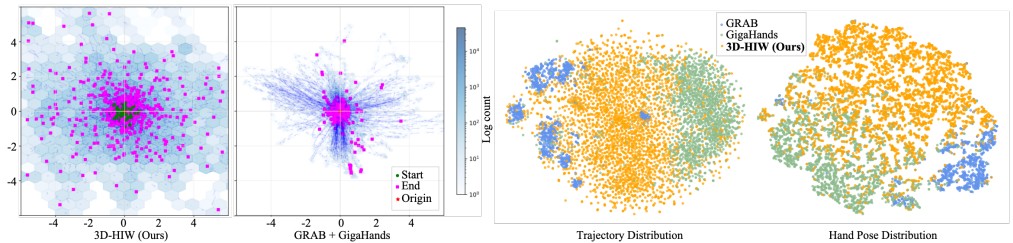

Figure 6: Comparison of our 3D-HIW dataset with existing datasets (GRAB, Gigahands). Left: 2D trajectory density plots show that our dataset covers a broader spatial range with more diverse start–end distributions. Right: t-SNE embeddings of trajectories and hand poses further highlight that our data spans a significantly wider distribution, capturing natural variability.

## 4.1 STRUCTURING HANDS INTO FINE-GRAINED TOKENS (SHIFT):

Standard VQ-VAE models struggle to capture the diversity and complexity of 'in-the-wild' hand motion, often resulting in limited reconstruction quality and generalization. To address this, we introduce SHIFT tokenizer that models trajectory and pose components using separate VQ-VAEs, while also disentangling left and right hands during encoding and decoding. This design choice is motivated by prior findings from Huang et al. (2025); Chen et al. (2024), where separating motion into different parts like

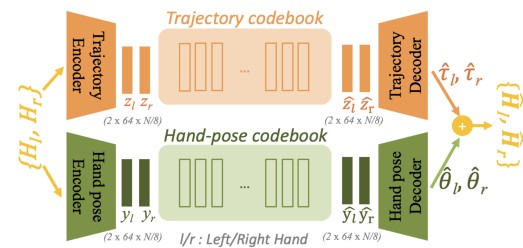

Figure 7: **SHIFT Tokenizer overview.**

hand, face, and objects shows improved performance. Our work extends this idea further by separating the motion into part-modality-specific granular components. Empirically, this formulation achieves stronger generalization and more faithful reconstructions (Table 4), even under high temporal compression (Figure 10). The hand motions are encoded using trajectory $E_\tau$ and hand pose $E_\theta$ encoders to produce $z_j \in \mathbb{R}^{d \times N/8}$ and $y_j \in \mathbb{R}^{d \times N/8}$ embeddings, where $d$ represents the dimension of the codebook latent space. The embeddings are quantized into $\hat{z}_j$ and $\hat{y}_j$ using nearest neighbor quantization (van den Oord et al., 2018). The trajectory $\hat{\tau}_j$ and hand pose $\hat{\theta}_j$ of the input sequence is reconstructed using the respective decoders $D_\tau$ and hand pose $D_\theta$, to get the final reconstructed motion $\hat{M} = (\hat{\tau}_j, \hat{\theta}_j)$ we train the encoder, decoder, and codebook simultaneously with the loss:

$$\mathcal{L}_{VQ} = \mathcal{L}_{rec}(M, \hat{M}) + \sum_{x \in X} \left( \|\text{sg}[x] - \hat{x}\|^2 + \beta \|x - \text{sg}[\hat{x}]\|^2 \right), \quad X = \{z_l, z_r, y_l, y_r\}, \quad (1)$$

where $\mathcal{L}_{rec}$ is an MSE reconstruction loss, sg is a stop gradient operation used to calculate the codebook loss, and the third part is a "commitment" loss with a trade-off $\beta$.

## 4.2 LLM FOR HAND-MOTION MODELLING:

Employing the part-modality decomposed tokenizer, a hand motion sequence $M_{1:N}$ can be mapped to discrete trajectory and pose tokens $\mathbf{z}_{1:T} = \{z_t\}_{t=1}^T$ and $\mathbf{y}_{1:T} = \{y_t\}_{t=1}^T$. We represent the motion tokens as sequences of indices $\mathbf{s}_{1:2T} = \{s_t\}_{t=1}^{2T}$, $s_t \in \mathbb{N}$, where each $s_t$ is drawn from the combined motion vocabulary space $V_m$, where trajectory and pose codebooks are stacked. When tokenized, the motion sequence is represented as an interleaved stream of trajectory and pose tokens. In practice, each motion token is written as a special symbol <motion_token{i}>. For brevity, we denote motion tokens as $\langle m \rangle$ and text tokens as $\langle t \rangle$.

For example, a sequence with $T$ trajectory $\langle m^{(\tau)} \rangle$ and pose tokens $\langle m^{(\theta)} \rangle$ is arranged as:

$$\langle \text{som} \rangle \langle m_1^{(\tau_L)} \rangle \langle m_1^{(\theta_L)} \rangle \langle m_1^{(\tau_R)} \rangle \langle m_1^{(\theta_R)} \rangle; \cdots; \langle m_T^{(\tau_L)} \rangle \langle m_T^{(\theta_L)} \rangle \langle m_T^{(\tau_R)} \rangle \langle m_T^{(\theta_R)} \rangle \langle \text{eom} \rangle. \quad (2)$$

To train the LLM, we build a unified text–motion space $V = V_t \cup V_m$, where $V_t$ is the text vocabulary. We include additional special tokens such as boundary markers (e.g., <som>, <eom>), which enable text-conditioned motion tasks to be represented in a consistent format. The model handles text-to-motion, motion-to-text, or joint captioning tasks in a unified manner. Given an input sequence

$X_s = \{x_k^s\}_{k=1}^K,\ x_j^s \in V$, it predicts the target sequence $X_t = \{x_i^t\}_{i=1}^L,\ x_i^t \in V$ autoregressively:

$$p_\theta(X_t \mid X_s) = \prod_{i=0}^{L-1} p_\theta\big(x_i^t \mid x_{<i}^t, X_s\big). \tag{3}$$

The training objective is:

$$\mathcal{L}_{LM} = -\sum_{i=0}^{L-1} \log p_\theta\big(x_i^t \mid x_{<i}^t, X_s\big). \tag{4}$$

**Pre-training Stage.**  We pre-train the language model on large-scale text and motion sequences using a cross-entropy loss on the next-token-prediction task and simple T2M and M2T tasks. This allows the model to capture natural language semantics and temporal dynamics of hand motions, similar to MotionGPT.

**Geometric-Refinement Stage.**  While token-level cross-entropy loss encourages correct next-token prediction, we find it does not guarantee that decoded motions are geometrically smooth or realistic. Prior works (Hong et al., 2024) address this by adding soft-blending-based regression losses during the pre-training stage. However, jointly applying soft-blending-based regression in pre-training conflicts with cross-entropy, as soft-blending favors smooth interpolations while CE enforces sharp token predictions, leading to modest performance improvements (Table 5). To address this, we adopt a Gumbel-Softmax parameterization, which enables discrete token selection while directly applying regression loss in motion space. This yields the joint training objective: $\mathcal{L} = \alpha\mathcal{L}_{LM} + \lambda\mathcal{L}_{rec}$, where $\mathcal{L}_{rec}$ ensures fidelity of the reconstructed hand motion. In addition, we also train the model on additional masked prediction tasks with $\alpha = 0$ to encourage the model to focus more on the reconstruction quality.

**Instruction Fine-tuning Stage.**  Finally, we perform instruction fine-tuning to enable the model to handle multiple tasks, including text-to-motion and motion-to-text. We adopt the multi-task prompt-based training strategy from MotionGPT, where the model is supervised on diverse instruction prompts. This stage improves generalization across different tasks and yields state-of-the-art performance on both synthesis and captioning benchmarks.

## 5 EXPERIMENTS

**Dataset**  We build our experiments on the proposed 3D-HIW hand motion dataset, which provides paired 3D hand motions and text descriptions of 32k real-world sequences. For training and evaluation, we partition the sequences into non-overlapping splits to avoid leakage between sets. Specifically, we allocate **80% for training** (26k sequences), **10% for validation** (3k), and **10% for testing** (3k).

**Evaluation Metrics:**  For **text-to-motion generation (T2M)**, we follow prior work Tevet et al. (2023); Guo et al. (2022) and report: *R Precision (RP3)* for text–motion matching, *MMD* for text and motion alignment in feature space, *KID* for distribution similarity, and *Diversity* for output variability and *Multimodality* for diversity from a single prompt. For **motion-to-text captioning**, we use standard language metrics (*BLEU4*, *BLEU1*, *Rouge-L*) along with *R Precision*. For **annotation quality**, we adopt GPT-Score following EgoHOD Hong et al. (2024). For **motion reconstruction**, we report *MPJPE*, *PA-MPJPE*, and *ACCEL* as in EgoLM Hong et al. (2024).

### 5.1 DATASET ANNOTATION

We evaluate the quality of our egocentric video-to-text annotations using GPT-Scores from the EgoHOD (Pei et al., 2025), which rate similarity to human-authored descriptions on a 0–10 scale (higher is better). Results are reported in Table 3. Compared to LaVILA (Zhao et al., 2023) and EgoHOD, our method achieves the highest GPT-Score (6.9), surpassing existing approaches by a clear margin. This confirms that our pipeline produces more faithful and higher-quality text annotations. To further analyze the role of our two-stage annotation pipeline, we ablate against two baselines: (i) **VILA-Naive**, which uses a single large prompt, and (ii) **VILA-Stage1**, which only uses the first-stage

Table 1: Comparison of various methods on RPrecision, MMDist, KID Mean, Diversity, and MultiModality. Lower is better for all metrics except RPrecision and Diversity.

| Method | RP3 ↑ | MMD ↓ | KID ↓ | Div → | MM ↑ |
|---|---|---|---|---|---|
| Ground Truth | 0.667 ± 0.004 | 1.903 ± 0.005 | | 3.964 ± 0.189 | |
| HumanMDM | 0.694 ± 0.005 | 1.971 ± 0.019 | 0.344 ± 0.02 | 3.824 ± 0.177 | 1.748 ± 0.069 |
| MotionGPT | 0.573 ± 0.009 | 2.183 ± 0.013 | 0.756 ± 0.03 | 3.642 ± 0.119 | 2.015 ± 0.095 |
| T2M-GPT | 0.683 ± 0.005 | 1.976 ± 0.011 | 0.431 ± 0.02 | 3.854 ± 0.130 | 1.892 ± 0.085 |
| **Ours** | **0.721 ± 0.004** | **1.765 ± 0.016** | **0.216 ± 0.02** | **3.865 ± 0.124** | 1.984 ± 0.084 |

outputs. Both underperform compared to our full pipeline, validating the importance of structured multi-stage prompting for robust annotation quality. We study motion quality of the 3D-HiW dataset with respect to different data-cleaning steps in Appendix C.1.

Table 2: Motion-to-text captioning quantitative results.

| Method | RP3 ↑ | B4 ↑ | B1 ↑ | RG ↑ |
|---|---|---|---|---|
| GT | 0.668 | | | |
| TM2T | 0.385 | 0.122 | 0.333 | 0.428 |
| MotionGPT | 0.407 | 0.132 | 0.345 | 0.439 |
| **Ours** | **0.571** | **0.181** | **0.420** | **0.472** |

Table 3: Evaluating text annotations using EgoHoD's GPT-Scores (0–10).

| Method | GPT-Score ↑ |
|---|---|
| LaVILA | 4.9 ± 0.3 |
| EgoHOD | 6.1 ± 0.4 |
| VILA-Naive | 5.5 ± 0.2 |
| VILA-Stage1 | 6.4 ± 0.5 |
| **Ours** | **6.9 ± 0.3** |

## 5.2 CLUTCH – TEXT-TO-MOTION GENERATION (T2M)

The text-to-motion task evaluates a model's ability to generate plausible hand motion sequences given natural language input. We benchmark CLUTCH against recent state-of-the-art baselines, including MotionGPT (Jiang et al., 2024), HumanMDM (Tevet et al., 2023), and T2MGPT (Zhang et al., 2023), retraining all models on our dataset for fairness. Results are reported in Table 1. Across all metrics, CLUTCH achieves consistent improvements over competing methods, suggesting that its unified modelling of language and hand motion provides stronger alignment than prior approaches. Qualitative results in Figure 8 further highlight CLUTCH's ability to generate multiple diverse yet semantically faithful motion trajectories from the same textual description.

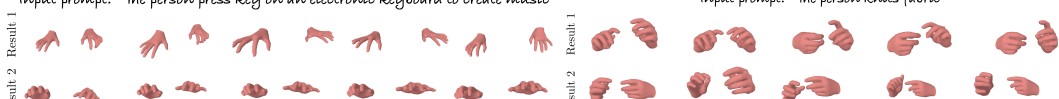

Figure 8: Qualitative results for text-to-motion synthesis.

## 5.3 CLUTCH – MOTION-TO-TEXT CAPTIONING (M2T)

The motion-to-text task involves generating text descriptions from novel 3D hand motions from the wild. To this end, we compare our method against MotionGPT and TM2T (Guo et al., 2022) and report the metrics in Table 2. From the results, we can infer that our method significantly outperforms the baselines on all the metrics. We show qualitative results of motion captioning in Figure 9.

## 5.4 ABLATIONS

**Effectiveness of the SHIFT tokenizer:** We compare our SHIFT with three baselines: MotionGPT's VQ-VAE, a standard VQ-VAE, and a part-decomposed variant (PD VQ-VAE) that disentangles left and right hands during encoding and decoding. As shown in Table 4, our model achieves the best overall performance, yielding the lowest MPJPE (45.94) and ACCEL (5.395), while also improving motion diversity. Moreover, Figure 10 illustrates that SHIFT handles temporal compression substantially better than the baseline VQ-VAEs, enabling LLM training under modest memory requirements (4 A100 GPU's vs 64 Tesla V100 and 32 NVIDIA A100 GPU's in MotionGPT and HoiGPT respectively). These results underscore the advantage of decomposing both body parts and modalities in VQ-VAE–based motion modelling. Additional experiments are presented in Appendix B.2.

**Impact of Geometric Refinement and Instruct-Fine Tuning:** Table 5 compares different training stages. Pre-training alone (row 1) provides a reasonable baseline, but performance remains limited. Instruction tuning (IFT) substantially improves results (row w/o GR), raising T2M RP3 from 0.53 to

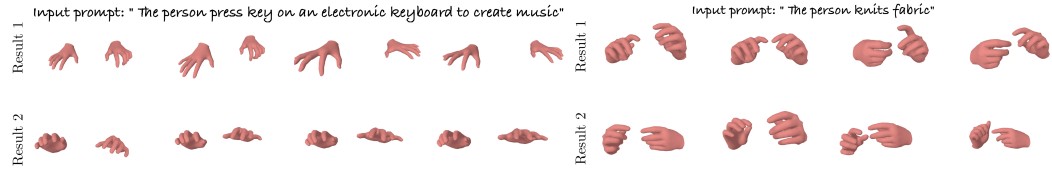

Figure 9: Motion-to-Text captioning results.

Table 4: Comparison of VQ-VAE configurations.

| Method | Num. / dim | MPJPE ↓ | ACCEL ↓ | Div → |
|---|---|---|---|---|
| GT | – | – | – | 3.964 |
| MotionGPT | 512 / 512 | 93.486 | 8.340 | 3.683 |
| Std. VQ-VAE | 4K / 64 | 93.258 | 7.771 | 3.450 |
| PD VQ-VAE | 4K / 64 | 95.266 | 7.500 | 3.647 |
| **Ours** | 4K / 64 | **45.944** | **5.395** | **3.747** |

Figure 10: VQ-VAE compression.

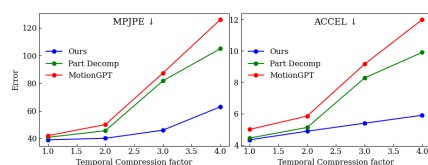

0.69 and M2T RP3 from 0.50 to 0.57. Adding geometric refinement (GR) further boosts alignment: the full model (PT+GR+IFT) achieves the lowest KID (0.216 vs. 0.297 w/o GR) and the highest RP3 scores (0.72 for T2M, 0.57 for M2T). This demonstrates that GR plays a key role in motion synthesis quality. In other words, IFT scales generalization, while GR makes that generalization meaningful by enforcing geometric alignment. The combination yields the best overall performance. Finally, we compare against the EgoLM Hong et al. (2024) soft-blending reconstruction loss (last row). While competitive, it is inferior to our approach, highlighting the benefits of explicit geometric refinement and Gumbel-Softmax–based reconstruction.

**Impact of Dataset Size:** Increasing the number of captioned sequences from 7K to 30K yields steady improvements in both text-to-motion (T2M) and motion-to-text (M2T). These results underline the importance of larger, more diverse training data for scalable in-the-wild hand motion modelling. For reference, we also provide our method trained on a combination Arctic and GRAB dataset.

**Impact of Language Model Size:** Table 7 reports the effect of scaling the backbone language model from T5-Small to T5-Large. As expected, larger models yield consistently better results on both T2M and M2T tasks. These results confirm that language model capacity plays a crucial role in enabling stronger generalization across modalities in both tasks.

Table 7: Impact of model size on the performance.

| Method | T2M | | M2T | |
|---|---|---|---|---|
| | RP3 ↑ | KID ↓ | RP3 ↑ | B4 ↑ |
| T5-Small (50M) | 0.545 | 0.732 | 0.292 | 0.089 |
| T5-Base (220M) | 0.721 | 0.216 | 0.571 | 0.181 |
| T5-Large (770M) | 0.733 | 0.092 | 0.578 | 0.192 |

## 6 CONCLUSION

To the best of our knowledge, CLUTCH is the first work to explore in-the-wild hand motion modelling. While effective, our approach still has limitations. We focus on hand motions, while leaving hand–object interactions for future exploration due to the current challenges of in-the-wild reconstruction. Further improvements may enhance fine-grained expressiveness in motion reconstructions and enable temporal segmentation of overlapping actions in egocentric sequences. Advancing along these directions could further improve dataset quality and model robustness.

Table 5: Impact of different LLM training stages. PT: Pre-training, GR: Geometry refinement, IFT: Instruct Fine-tuning.

| Method | T2M | | M2T | |
|---|---|---|---|---|
| | RP3 ↑ | KID ↓ | RP3 ↑ | B4 ↑ |
| 1 = PT | 0.533 | 0.349 | 0.501 | 0.148 |
| w/o GR (1+IFT) | 0.690 | 0.297 | 0.568 | 0.173 |
| PT + GR + IFT | **0.721** | **0.216** | **0.571** | **0.181** |
| EgoLM setup | 0.705 | 0.263 | 0.570 | 0.171 |

Table 6: Performance scaling with increased training data (7K, 15K, 30K samples). *Cap. data:* Artic+GRAB.

| Method | T2M | | M2T | |
|---|---|---|---|---|
| | RP3 ↑ | KID ↓ | RP3 ↑ | B4 ↑ |
| *Cap. data* | 0.097 | 1.970 | 0.083 | 0.004 |
| 7K | 0.513 | 0.860 | 0.247 | 0.092 |
| 15K | 0.637 | 0.672 | 0.396 | 0.139 |
| **30K** | **0.721** | **0.216** | **0.571** | **0.181** |

Despite these challenges, CLUTCH makes important progress towards scalable, natural hand motion synthesis. To this end, we introduce a novel data annotation pipeline, a dataset, and a part-modality decomposed VQ-VAE for in-the-wild hand motion modelling. Through detailed experiments, we demonstrate that CLUTCH outperforms existing diffusion and LLM models on the in-the-wild hand motion modelling task. Looking ahead, we believe combining in-the-wild motions with controlled datasets, and extending to hand–object interactions can unlock new downstream applications in behavioral AI, allowing us to eventually build embodied avatars capable of fine-grained high-fidelity interactions with their environments.

## 7  ACKNOWLEDGMENTS:

Justus Thies is supported by the DFG Excellence Strategy— EXC-3057 (”Reasonable Artificial Intelligence“, Project No. 533677015) and the project is co-funded by the European Union (ERC, Lemo, 101162081). Views and opinions expressed are, however, those of the author(s) only and do not necessarily reflect those of the European Union or the European Research Council. Neither the European Union nor the granting authority can be held responsible for them. Gerard Pons-Moll is supported by the Carl Zeiss Foundation, Amazon, MPI Science Hub and the Deutsche Forschungs-gemeinschaft (DFG, German Research Foundation) under Grant No. 409792180 (Emmy Noether Programme, project: Real Virtual Humans). He is further supported by the German Federal Ministry of Education and Research (BMBF) through the Tübingen AI Center (FKZ: 01IS18039A). Gerard Pons-Moll is a member of the DFG Cluster of Excellence “Machine Learning – New Perspectives for Science” (EXC 2064/1, Project No. 390727645). Further, we also thank B. Kabadayi, M. Diomataris, S. Tripathi and P. Mayilvahanan for valuable discussions.

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

# A GUMBEL-SOFTMAX MOTION DECODING.

Given an input sequence $X_s$, the LLM outputs a full vocabulary logit tensor $f_\theta(X_s) \in \mathbb{R}^{T \times |V|}$, where $|V|$ is the joint (text + motion) vocabulary. For motion decoding, we extract *only the logits corresponding to the motion–token subspace* $V_m \subset V$. This slicing is expressed as:

$$\ell_{1:T} = f_\theta(X_s)_{1:T, V_m},$$

where $\ell_t \in \mathbb{R}^K$ and $K = |V_m|$ is the size of the motion–token vocabulary. This corresponds exactly to selecting the motion–token logit channels from the full output tensor.

The extracted motion logits are then converted into a categorical representation through a Gumbel–Softmax operator (Jang et al., 2017):

$$\tilde{Z}_{1:T} = Gumbel(\ell_{1:T}, \tau).$$

The continuous 3D hand–motion sequence is reconstructed by decoding this Gumbel–Softmax motion representation using the SHIFT decoder:

$$\hat{M}_{1:T} = \mathcal{D}_\tau, \mathcal{D}_\theta(\tilde{Z}_{1:T}),$$

where $\mathcal{D}_\tau$ denotes trajectory decoder parameters and $\mathcal{D}_\theta$ the hand-pose decoder parameters.

**Reconstruction Loss.**  To refine geometric fidelity, we combine the language–modeling loss $\mathcal{L}_{LM}$ with a reconstruction loss computed in continuous motion space:

$$\mathcal{L}_{rec} = \frac{1}{T} \sum_{t=1}^{T} \left\| \hat{M}_t - M_t \right\|_2^2.$$

The final objective is:

$$\mathcal{L} = \alpha \mathcal{L}_{LM} + \lambda \mathcal{L}_{rec}.$$

# B ADDITIONAL EXPERIMENTS

**Implementation details:** In our experiments, we use two VQ-VAE models with 4096 codebook entries of 64 dimensions each. The compression rate of the VQ-VAE is 8, i.e., the encoder compresses 8 temporal frames into a single code. The motion tokenizer is trained for 2000 epochs using the Adam optimizer with a learning rate of $2e^{-4}$. We employ the 220M-parameter Flan-T5-Base (Roberts et al., 2022) as our language model. The model is pre-trained, geometry-refined, and fine-tuned for $300/50/200$ epochs with learning rates of $2e^{-4}/1e^{-5}/2e^{-5}$, respectively. Experimental results are reported with a 95% confidence interval, computed from 20 repeated runs to ensure statistical significance. All models are trained on 4 NVIDIA A100 GPUs with 80GB memory each.

## B.1 EFFECTIVENESS OF TEXT-ANNOTATION TYPE:

We evaluate how different annotation types affect LLM performance, using high-level (HA), fine-grained (DA), and combined (HA+DA) annotations (Table 8). Using only high-level (HA) or fine-grained (DA) annotations yields moderate performance (e.g., T2M RP3 = 0.551 and 0.462). Combining both (HA+DA) yields the best results across metrics (T2M RP3 = 0.721, M2T RP3 = 0.571), underscoring their complementarity for robust text–motion learning.

Table 8: Effect of different types annotation on Text-to-Motion task performance. HA: High-level annotation, DA: Fine-grained Annotation.

| Method | T2M RP3 ↑ | T2M KID ↓ | M2T RP3 ↑ | M2T B4 ↑ |
|---|---|---|---|---|
| Ours (HA) | 0.551 | 0.148 | 0.496 | 0.153 |
| Ours (DA) | 0.462 | 0.192 | 0.489 | 0.114 |
| **Ours(HA+DA)** | 0.721 | 0.216 | 0.571 | 0.181 |

## B.2 TOKENIZER ANALYSIS:

We provide additional comparisons of our decomposed VQ-VAE (SHIFT) against several baselines to further highlight the impact of model design choices. As reported in Table 9, our formulation

consistently achieves the lowest reconstruction error, with MPJPE reduced to 45.94 and ACCEL to 5.395, while preserving motion diversity. Further, We visualize the effect of temporal compression in Figure 10. Whereas standard VQ-VAEs degrade rapidly as the compression factor increases, our decomposition into trajectory and pose codebooks maintains reconstruction quality even at high compression rates. This property is especially important for scaling large language models to motion, as it reduces the effective sequence length and enables training under more modest compute budgets. In practice, our model requires only 4 NVIDIA A100 GPUs for training, compared to the 64 Tesla V100 GPUs used in MotionGPT and 32 A100 GPUs in HOIGPT. These extended experiments confirm that decomposing both modalities (trajectory vs. pose) and body parts (left vs. right hand) is a crucial factor for stable, scalable motion modeling.

Table 9: **VQVAE analysis - Extended version**

| Method | Num. / dim | MPJPE $\downarrow$ | ACCEL $\downarrow$ | Div $\rightarrow$ |
|---|---|---|---|---|
| GT | | | | 3.964 |
| MotionGPT | 512 / 512 | 93.486 | 8.340 | 3.683 |
| Std. VQ-VAE | 4K / 64 | 93.258 | 7.771 | 3.450 |
| Std VQ-VAE | 8K / 64 | 92.150 | 7.859 | 3.539 |
| Std VQ-VAE | 4K / 256 | 93.045 | 8.014 | 3.647 |
| PD VQ-VAE | 4K / 64 | 95.266 | 7.500 | 3.647 |
| PD VQVAE | 8K / 64 | 92.052 | 7.369 | 3.581 |
| PD VQVAE | 4K / 256 | 97.289 | 7.616 | 3.357 |
| **Ours** | 4K / 64 | **45.944** | **5.395** | **3.747** |

### B.3 RESULTS ON PUBLIC DATASETS:

To further assess the capability of our method, we follow the dataset protocol of HOIGPT(Huang et al., 2025) and train our model and all baselines on a publicly available captured dataset composed of ARCTIC(Fan et al., 2023) and GRAB (Taheri et al., 2020), covering 5.1K / 0.5K / 0.5K sequences for training, validation, and testing. We evaluate performance on the Text2Motion (T2M) and Motion2Text (M2T) tasks using the metrics described in Section 5, and we report the results in Table 10 and Table 11.

As shown in the tables, our method consistently outperforms prior approaches across both tasks. In T2M, our model achieves the highest R-Precision (0.492), the lowest MMDist among generative models (3.008), and competitive KID scores, while also providing substantially better multimodality than MotionGPT(Jiang et al., 2024) and T2MGPT(Zhang et al., 2023). Notably, HumanMDM (Tevet et al., 2023), a diffusion-based model, tends to generate visually smooth but less semantically aligned motions, which is reflected in its lower R-Precision and higher MMDist under this reduced-data regime. In M2T, our method again achieves the best performance across all major metrics, indicating stronger bidirectional grounding between motion and language compared to MotionGPT and TM2T. Although our model is explicitly designed for in-the-wild hand-motion modeling, it nonetheless generalizes effectively to controlled HOI datasets, demonstrating the strength and versatility of the learned representation.

### B.4 SENSITIVITY ANALYSIS OF THE LM AND RECONSTRUCTION LOSSES

We conducted a full $\alpha/\lambda$ sensitivity sweep to study the effect of balancing the language-modeling loss and the reconstruction loss. The results are presented in Table 12. We observe a consistent trend:

| Method | RP3 $\uparrow$ | MMDist $\downarrow$ | KID $\downarrow$ | Diversity $\rightarrow$ | MultiModality $\uparrow$ |
|---|---|---|---|---|---|
| Ground Truth | 0.525 | 2.763 | – | 4.581 | – |
| HumanMDM | 0.429 | 4.047 | 0.0107 | 4.915 | 2.567 |
| MotionGPT | 0.371 | 3.609 | 0.0409 | 3.315 | 1.955 |
| T2MGPT | 0.407 | 3.761 | 0.0773 | 4.956 | 1.658 |
| **Ours** | 0.492 | 3.008 | 0.0144 | 3.811 | 2.393 |

Table 10: T2M evaluation results on ARCTIC+GRAB.

| Method | RP3 ↑ | Bleu4 ↑ | Bleu1 ↑ | ROUGE_L ↑ |
|---|---|---|---|---|
| TM2T | 0.3519 | 0.1815 | 0.2245 | 0.5174 |
| MotionGPT | 0.4262 | 0.2158 | 0.5167 | 0.5278 |
| **Ours** | 0.4601 | 0.2341 | 0.5732 | 0.5822 |

Table 11: M2T evaluation results on ARCTIC+GRAB.

| | | T2M | | M2T | |
|---|---|---|---|---|---|
| **LM ($\alpha$)** | **Rec ($\lambda$)** | **RP3 ↑** | **KID ↓** | **RP3 ↑** | **Bleu4 ↑** |
| GT | | 0.671 | – | 0.667 | – |
| **0** | **1** | **0.413** | **0.886** | **0.099** | **0.021** |
| 0.1 | 0.9 | 0.498 | 0.725 | 0.357 | 0.077 |
| 0.25 | 0.75 | 0.522 | 0.335 | 0.403 | 0.116 |
| **0.5** | **0.5** | **0.721** | **0.216** | **0.571** | **0.181** |
| 0.75 | 0.25 | 0.712 | 0.234 | 0.544 | 0.172 |
| 0.9 | 0.1 | 0.708 | 0.289 | 0.543 | 0.171 |
| **1** | **0** | **0.690** | **0.297** | **0.568** | **0.173** |

Table 12: Sensitivity study of the LM loss weight $\alpha$ and reconstruction loss weight $\lambda$. Left: M2T performance (RP3, KID). Right: T2M performance (RP3, Bleu4). GT: Ground Truth

large $\lambda$ (low $\alpha$) smooths the motion but affects semantic alignment, while large $\alpha$ (low $\lambda$) sharpens token prediction but increases geometric artifacts, reflected in higher KID scores. The balanced setting of $\alpha = 0.5, \lambda = 0.5$ delivers the best overall performance across both M2T (RP3 = 0.721, KID = 0.216) and T2M (RP3 = 0.571, Bleu4 = 0.181).

When $\lambda$ is high (i.e., the reconstruction loss dominates), the model struggles to capture the overall distribution, highlighting the importance of the LM loss for maintaining semantic alignment. Conversely, when $\alpha$ is too high, the model predicts sharper discrete tokens but exhibits poorer geometric realism. These findings confirm that a balanced loss weighting is essential for high-quality motion generation.

| Method | Rating (1-5) |
|---|---|
| A = Random | 1.106 |
| B = Our annotation pipeline | 4.244 |
| C = Human annotation | 4.673 |

| Method | Rating (1-5) |
|---|---|
| A = Random motion | 1.375 |
| B = Without filters | 2.434 |
| C = **Final-cleaned** | **4.133** |

Table 14: User study results. Left: annotation quality ratings. Right: motion quality ratings. **Rating: 1 = Low, 5 = Best**

## C  3D HANDS IN THE WILD (3D-HIW) DATASET - EXTENSION:

### C.1  DATASET ANALYSIS - CONTINUATION:

To extract 3D hand motion reconstructions from egocentric videos, we first process high-level text descriptions from the EgoVid5M dataset to identify sequences involving human presence, particularly those where humans interact with objects. We then cluster these descriptions into scene-level categories (e.g., crafting, repair) and sample uniformly across clusters to mitigate the overrepresentation of cooking activities. We

Table 13: **Dataset design choices evaluation**

| Method | RP3 ↑ | MMD ↓ |
|---|---|---|
| w/o. both hands filter | 0.178 | 3.819 |
| w/o. accl. | 0.422 | 2.653 |
| w/o. temp smooth. | 0.511 | 2.249 |
| w/o. verifier filter. | 0.553 | 2.019 |
| Ours (final) | 0.666 | 1.903 |

also study the impact of filter with respect to motion quality in Table 13, where we ablate key components of our cleaning pipeline. Removing filters (e.g., hand visibility checks, acceleration constraints, or temporal smoothing) significantly degrades R-Precision and increases motion noise. Further, we analyze the distribution of top-35 verbs and nouns in our dataset which is presented in Figure 11.

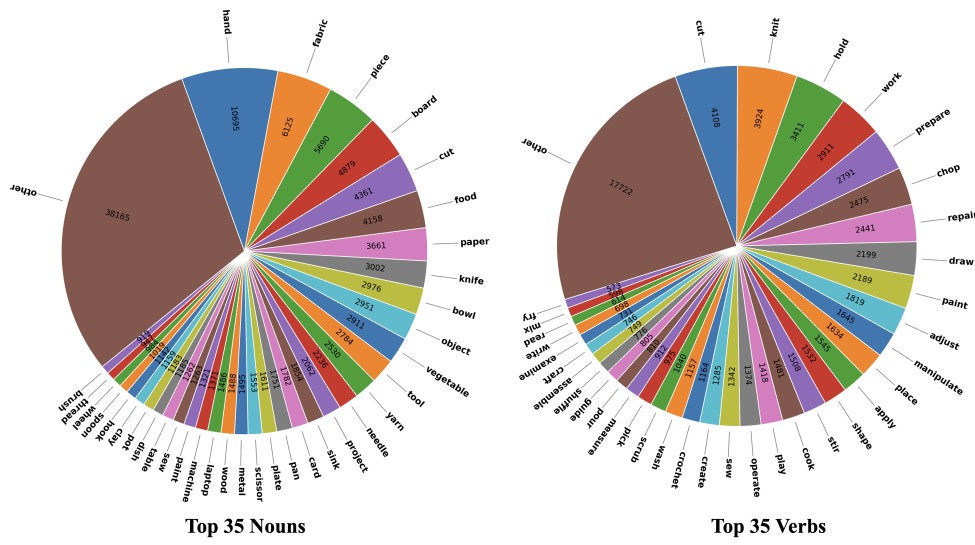

Figure 11: **Top-N Verb and Nouns:** We present the distribution of top-35 verbs and nouns in the '3D Hands in the wild' (3DHiW) dataset

### C.2  PERCEPTUAL USER-STUDY:

**Motion Reconstruction:** We conducted an additional MTurk user study to assess the perceptual quality of our reconstructed hand motions. Workers were shown the input egocentric video alongside two rendered 3D hand-motion reconstructions (front and back views), and were asked to rate on a 1-5 Likert scale how realistic the 3D motion appeared and how well it matched the motion in the

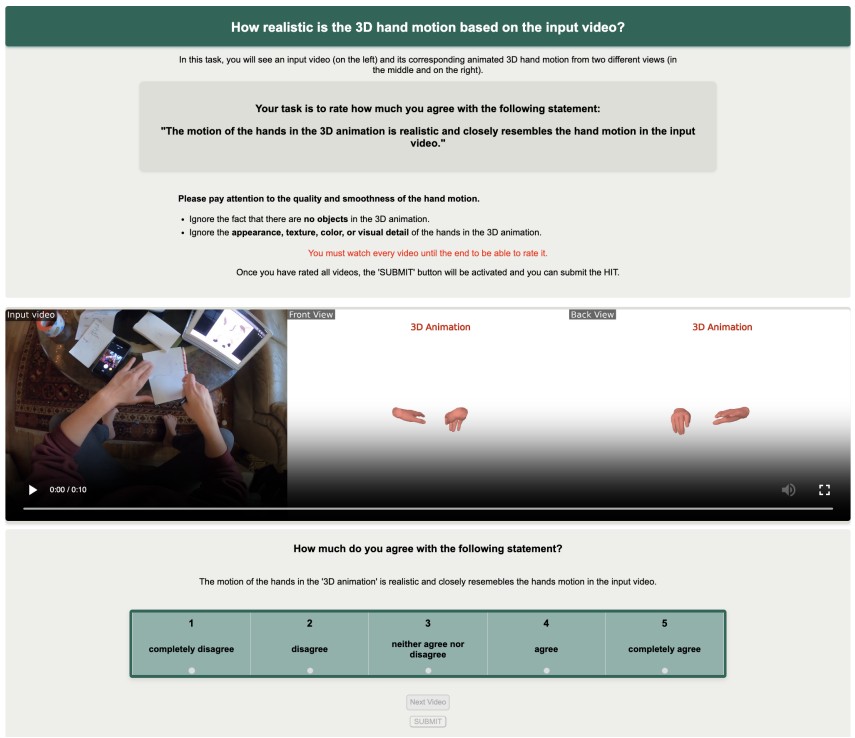

Figure 12: MTurk interface used for the motion user study.

video. We evaluate three categories: (A) random motions sampled from unrelated sequences, (B) our reconstruction without filtering, and (C) our final filtered reconstruction. From Table 14, users overwhelmingly preferred our final reconstruction (4.133) compared to the unfiltered version (2.434) and the random baseline (1.375). We restricted participation to experienced MTurk workers (>5000 HITs, ≥98% approval rate) and collected ratings on 65 sampled videos, with each video evaluated by 25 unique workers, resulting in a total of 1,625 judgments. The marked improvement from (B) to (C) confirms that our filtering pipeline substantially enhances motion quality. The MTurk user-study interface is presented in the Figure 12.

**Text annotation:** In addition, we conducted a human evaluation of the generated annotations using an MTurk study that mirrors the setup described above. Workers were shown an input egocentric video together with a candidate text description, and were asked to rate on a 1–5 Likert scale how much they agreed with the statement: "The text accurately describes the hand motion in the input video." We evaluate three categories: (A) a random annotation sampled from human annotation's, (B) our generated annotation, and (C) the corresponding human-written annotation. As reported in Table 14, random annotations received very low scores (1.106), confirming that workers reliably detect mismatched or incorrect text. Our generated annotations achieved a high rating of 4.244, which is close to the human-written descriptions (4.673). This strong alignment indicates that our automated annotation pipeline produces realistic and human-quality motion descriptions that accurately reflect the hand motions in the video. The MTurk interface used for this annotation study is shown in Figure 13.

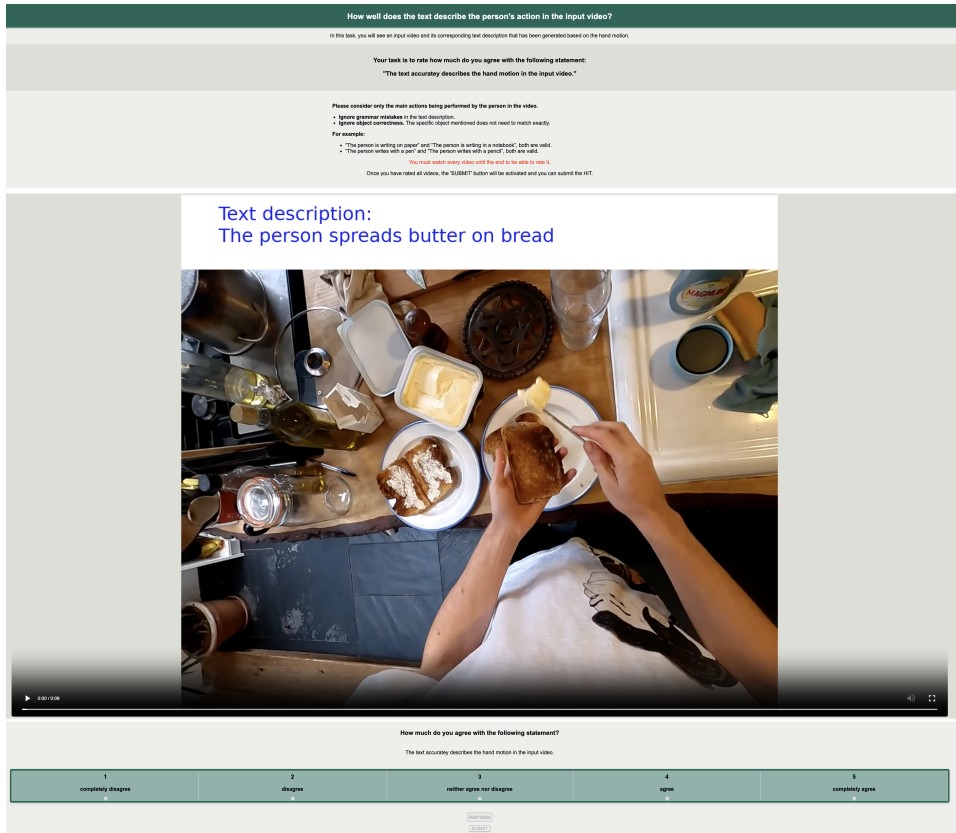

Figure 13: MTurk interface used for the text annotation user study.

## C.3 TEXT ANNOTATION PROMPTS:

.

Here, we give further details of the prompts introduced in Section 3.1 and Figures 3 and 4. In order to give the reader a better understanding of what is requested in the prompts, we give simplified (i.e. natural-language-based) prompt summaries in Figure 14. The actual exact prompts passed into the annotating LLM contain more formal language as well as a strict JSON output specification (following the example of Shorten et al. (2024)). The final prompts of both stages are given in Figure 15.

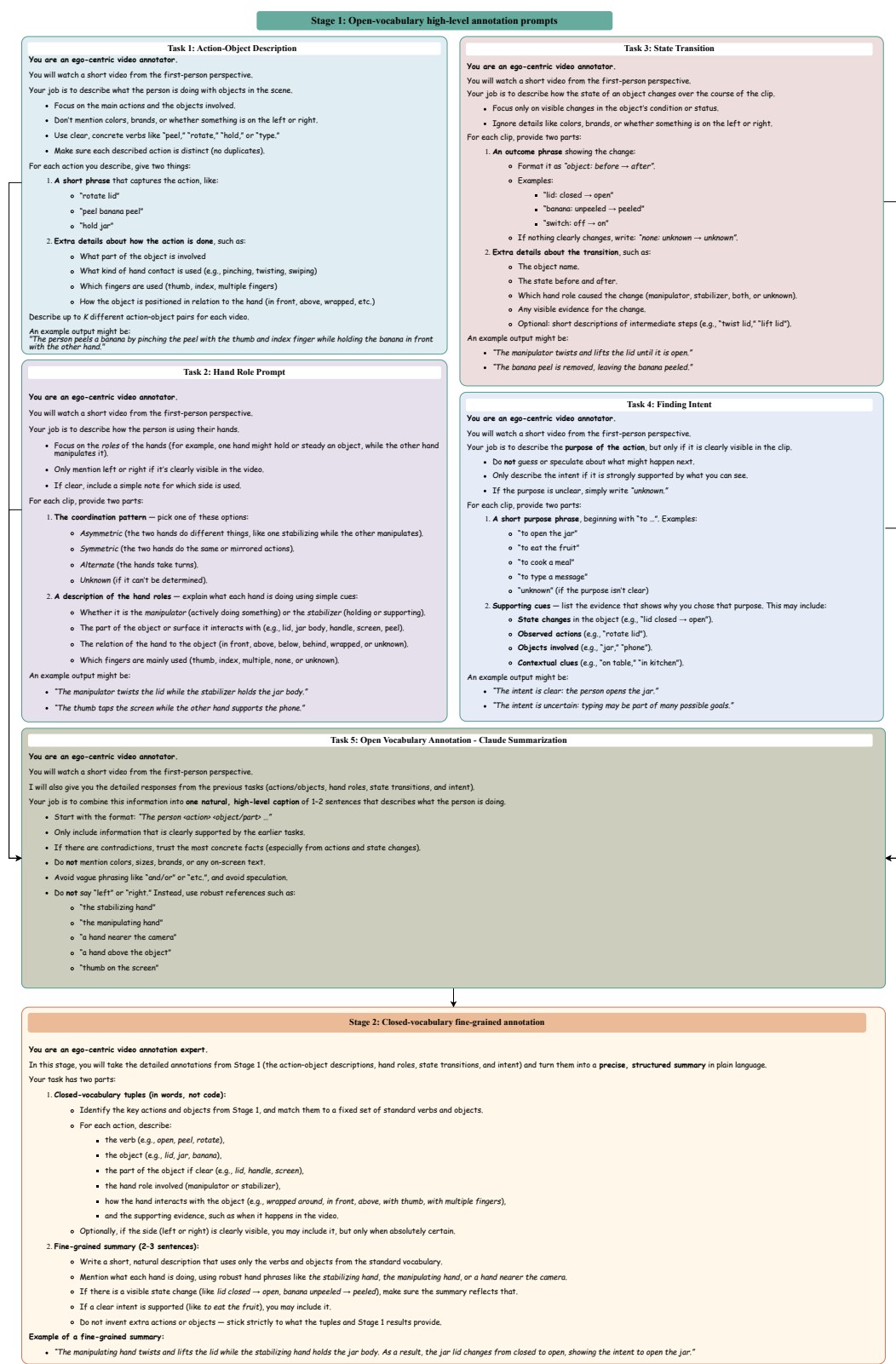

Figure 14: **Simplified natural language prompt summaries.** *First stage (top):* First 4 tasks are used for PCoT, and Task 5 is the open vocabulary summarization of the output of the first 4 tasks. *Second stage (bottom)* is used for the final closed-vocabulary fine-grained annotation generations.

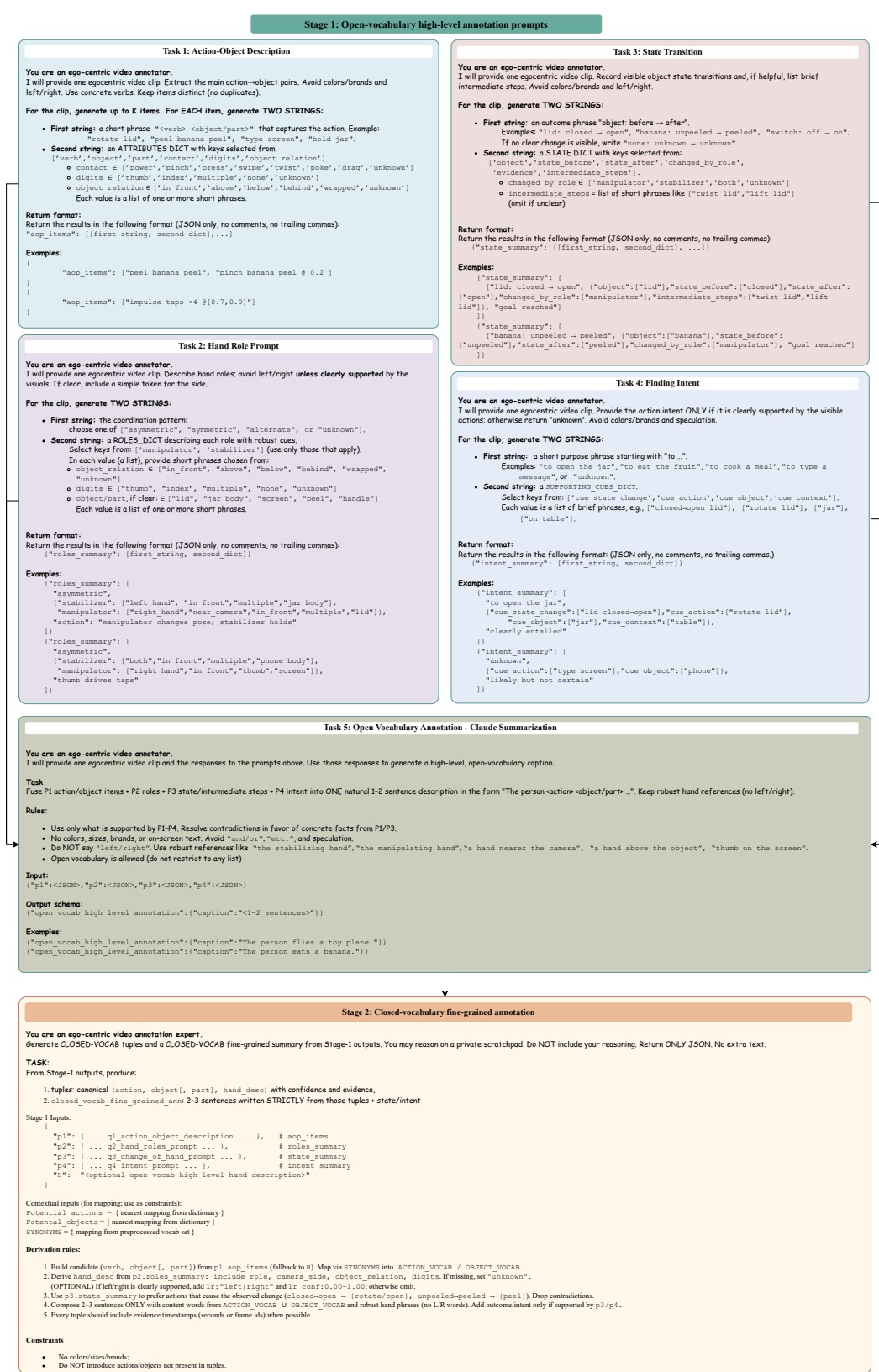

Figure 15: **The exact formal prompts** used in the data annotation pipeline. *First stage (top):* First 4 tasks are used for PCoT, and Task 5 is the open vocabulary summarization of the output of the first 4 tasks. *Second stage (bottom)* is used for the final closed-vocabulary fine-grained annotation generations. The prompts were designed following Shorten et al. (2024).

# D    RELATED WORKS

**Discussion:**   In contrast to prior work based on controlled mocap datasets or single-codebook tokenizers, we contribute the first in-the-wild 3D hand motion dataset with large-scale semantic annotations, a part-modality decomposed tokenizer for robust hand representation, and a geometry-aligned LLM training strategy. Together, these contributions enable CLUTCH to synthesize natural, diverse, and semantically consistent hand motions in unconstrained real-world settings.

## D.1    MOTION DATASETS / ANNOTATION

**Motion Datasets:** Existing motion datasets provide a foundation for body-level modelling but remain limited for hands. AMASS (Mahmood et al., 2019) aggregates mocap sequences, while GRAB, ARCTIC, H2O, DexYCB (Chao et al., 2021), and OakInk (Zhan et al., 2024; Yang et al., 2022) offer detailed 3D hand–object interactions. More recently, Gigahands (Fu et al., 2025) introduced a large dataset of 15K hand motion sequences with diverse actions and objects. However, these datasets are costly to collect, restricted to controlled studio settings, and cover only narrow action sets. Large-scale egocentric datasets such as Ego4D (Grauman et al., 2022) and EgoVid5M (Wang et al., 2024) capture diverse real-world activities, but lack accurate 3D hand reconstructions with semantic labels. This gap has so far prevented hand motion modelling from benefiting from large-scale training methods that have driven rapid advances in vision and language.

**Egocentric motion captioning:** Recent advances in egocentric video understanding have leveraged natural language for supervision, moving beyond classic action recognition tasks. LaViLa (Zhao et al., 2023), HOD (Pei et al., 2025), and EgoLM (Hong et al., 2024) are closest to our work on egocentric video to motion captioning. LaViLa and EgoLM leverage large language models (LLMs) to generate dense narrations for videos, while HOD augments these narrations by integrating detected hand–object trajectories with motion cues to produce semantically richer descriptions. In contrast, our method introduces a two-stage annotation pipeline: high-level open-vocabulary reasoning via parallel chain-of-thought prompting, followed by closed-vocabulary fine-grained grounding. This design reduces hallucinations, improves consistency, and yields scalable annotations tailored for text-to-motion modelling.

## D.2    MOTION MODELLING

**Full-body and Gesture Motion modelling:** Research in motion generation has largely focused on full-body and gesture synthesis (Guo et al., 2024; Liu et al., 2023; Zhang et al., 2023; Jiang et al., 2025; Wang et al., 2023; Shafir et al., 2023; Xie et al., 2023; Karunratanakul et al., 2023; Zhang et al., 2025c; 2023; Athanasiou et al., 2024; Chi et al., 2024; Chen et al., 2024; Liu et al., 2024; Habibie et al., 2024). Recent models, such as MDM Tevet et al. (2023) and MotionGPT Jiang et al. (2024), leverage transformer-based architectures and large-scale motion datasets to generate realistic human movements. Further, (Chen et al., 2024) built an multi-modal language models to unify the verbal and non-verbal 3D human motions. These approaches demonstrate strong performance on body-level actions but are primarily trained on controlled studio data, limiting their ability to generalize to fine-grained, unconstrained hand dynamics. While effective for large-scale gestures or locomotion, they fall short in modelling the nuanced variability of everyday hand behaviors.

**3D Hand-motion modelling:** A smaller body of work explicitly targets 3D hand motion modelling, where hands are modelled using MANO (Romero et al., 2017) and objects as 3D meshes. Recent works such as HOIGPT (Huang et al., 2025), and other hand-object interaction models (Christen et al., 2024; Cha et al., 2024; Li et al., 2025b; Ghosh et al., 2023) aim to capture fine hand-object interaction. However, they rely on high-quality mocap datasets such as GRAB Taheri et al. (2020), ARCTIC (Fan et al., 2023), and H2O (Kwon et al., 2021), which are limited in scale and diversity. Consequently, current hand motion models are often limited to narrow distributions of scripted actions.

**LLMs for motion modelling:** Large language models have recently been adapted for motion generation, leveraging their strengths in sequence modelling and cross-modal alignment. Works such as (Jiang et al., 2024; Huang et al., 2025; Chen et al., 2024) treat motion tokens as text-like symbols, enabling pretrained LLMs to transfer to motion tasks. While promising, these methods are limited by small-scale datasets and training objectives that emphasize token prediction accuracy rather than reconstruction fidelity. EgoLM (Hong et al., 2024) addresses this by introducing soft-linear blending

regression losses during pretraining, improving text–motion alignment. However, such regression objectives conflict with cross-entropy: blending encourages smooth interpolations, whereas CE enforces sharp token choices, leading to ambiguous representations and reduced generalization. Our approach extends this line of work with a geometry-alignment stage after pretraining, where Gumbel-Softmax sampling and hand motion reconstruction losses guide the LLM toward motions that are both semantically grounded and geometrically consistent.

**VQVAE as motion-prior:** Recent approaches discretize motion using VQ-VAE tokenizers, enabling motion to be represented in a language-like manner. Works such as (Jiang et al., 2024; Zhang et al., 2023) show that modelling motion as a sequence of tokens facilitates cross-modal learning with text. However, standard single-codebook tokenizers struggle to capture the multimodal nature of motion, where both trajectories and poses of different body parts must be jointly encoded. To address this, (Yi et al., 2023) introduce compositional codebooks for hand and face motion, while (Huang et al., 2025) employ separate codebooks for hand and object motion. Similarly, (Chen et al., 2024) decompose body parts into individual codebooks, each modeled independently. (Wang et al., 2025) further explore scaling strategies for codebooks to improve motion representation capacity. Building on these ideas, our formulation extends compositional quantization by introducing distinct codebooks for trajectories and hand poses, and further disentangling left and right hands during encoding and decoding. This design improves efficiency and generalization under higher temporal compression, while providing finer-grained control over multimodal hand motion generation compared to prior works.

