# OpenReview forum: "CLUTCH: Contextualized Language model for Unlocking Text-Conditioned Hand motion modelling in the wild"
_ICLR.cc/2026/Conference — ICLR 2026 Poster_

### Official Review · Reviewer_5Tyz · 2025-10-31

**Soundness:** 3
**Presentation:** 3
**Contribution:** 3
**Rating:** 4
**Confidence:** 3

**Summary:**

This paper addresses the problem of text-conditioned hand motion generation in unconstrained "in-the-wild" settings. The authors make three primary contributions: (1) a novel data annotation pipeline combining VLMs and 3D hand trackers to create the 3D-HIW dataset with 32K hand motion sequences and aligned text descriptions, (2) SHIFT, a part-modality decomposed VQ-VAE tokenizer that separately encodes trajectory and pose for left and right hands, and (3) CLUTCH, an LLM-based system featuring a geometric refinement training stage that applies reconstruction losses directly to decoded motion parameters. Experiments demonstrate improvements over baselines including HumanMDM, MotionGPT, and T2M-GPT on both text-to-motion and motion-to-text tasks.

**Strengths:**

1. The focus on in-the-wild hand motion generation addresses a significant gap in the literature, moving beyond studio-captured datasets with limited diversity.
2. The 3D-HIW dataset with 32K sequences represents approximately 10× the scale of GRAB and ARCTIC, offering substantially greater diversity in actions (1045 verbs, 1355 objects) and scenarios.
3. The two-stage approach using Parallel Chain-of-Thought prompting followed by closed-vocabulary refinement is well-motivated and demonstrates improved GPT-scores (6.9) compared to existing methods.

**Weaknesses:**

1. Dataset quality concerns:
- The reliance on HaWor for 3D reconstruction may introduce systematic errors or artifacts that propagate through the entire pipeline.
- The filtering criteria (80% hand visibility, acceleration thresholds) may introduce biases toward certain types of motions.
- No human evaluation or validation of the reconstructed 3D motions is provided.

2. Geometric refinement stage lacks clarity:
- The Gumbel-Softmax formulation is mentioned but not detailed.
- The balance between cross-entropy and reconstruction loss (α, λ) appears critical but hyperparameter sensitivity is not thoroughly analyzed.
- The comparison to EgoLM's soft-blending approach (Table 5) shows only marginal improvements, raising questions about whether the added complexity is justified.
3. The paper primarily compares against methods designed for full-body motion (HumanMDM) or general motion (MotionGPT). Comparisons with recent hand-specific methods or SOTA human motion generation method (MoMask) are missing.

**Questions:**

1. Have you conducted human evaluation of the generated annotations? What is the agreement between your automated annotations and human-written descriptions?
2. How does the model perform on motions significantly different from the training distribution? Can it generate novel compositions of actions?

---

> ### Author Response · Authors · 2025-11-24
> **Reply to reviewer 5Tyz (1/3)**
>
> We thank you for your helpful comments and feedback. We also appreciate the reviewer’s positive assessment of our contributions—specifically, the focus on **in-the-wild hand-motion generation**, the **scale and diversity of our 3D-HIW dataset** (32K sequences, ~10× GRAB/ARCTIC), and the effectiveness of our two-stage annotation pipeline, which achieves strong GPT-Scores. We address each of your questions and concerns below.
>
> **[W1- Motion quality]** :
> We understand the reviewer’s concern regarding the quality of the motion reconstruction. Based on the suggestion, we conducted an additional user study and report the results in **Appendix section.C.2**. In the following, we address your questions regarding data quality point by point.
>
> * **[W1a  HaWor and its systematic errors ]**: We employ HaWor[1] for motion reconstruction from egocentric videos. In our experiments, we observe three common systematic errors: (1) cases where the hands are not visible in most frames, (2) sudden hand motions causing motion blur, leading to jerky or chunked transitions, and (3) tracking noise resulting in jitter.  To address issues (1) and (2), we introduce three heuristic filters: (a) We filter out sequences, where 80% of both hands are not visible; (b) and (c) acceleration filters on translation and rotation to detect sudden jerky motions. Finally, to handle jitter, we apply a Savitzky–Golay filter followed by a Gaussian filter.  We qualitatively evaluate the motion reconstruction and present the results in **Table 10 in the appendix**, where we observe substantial improvements in RP3 (from 0.178 to 0.666) and MMD (from 3.819 to 1.903).
>
> * **[W1b: Bias towards to certain types of motion]**: We interpret the reviewer’s concern as referring to a potential bias toward bimanual motions due to our 80% hand-visibility criterion. We would like to clarify that this bias does not stem from the tracking or filtering scheme itself, but from the nature of egocentric video: when one hand is fully out of view, HaWor cannot reconstruct it, resulting in sequences where the missing hand either appears floating or produces physically implausible trajectories. Including such sequences would introduce systematic artifacts into the dataset. A straightforward way to expand the coverage of single-hand activities is to collect sequences where both hands remain visible but only one performs the action. Our dataset already contains several such examples (e.g., pouring from a bottle, stirring a pot), and we plan to increase this proportion in future extensions. If the reviewer is referring to another specific type of bias, we would be glad to receive clarification and address it to the best of our ability.
>
> * **[W1c: Motion Reconstruction User-Study]**: Following the reviewer’s suggestion, we conducted an additional MTurk user-study to assess the perceptual quality of our reconstructed hand motions. The workers are shown the input egocentric video alongside two rendered 3D hand-motion animations (front and back views) and rated, on a 1-5 Likert scale, how realistic the 3D motion was and how closely it resembled the motion in the video. We evaluated three setups: (A) random motions sampled from unrelated sequences, (B) our reconstruction without filters, and (C) our final filtered reconstruction. As we can see from table below, **users strongly preferred our final reconstruction (4.133) compared to the unfiltered version (2.434) and the random baseline (1.375)**. We restricted participation to experienced Turkers (>5000 HITs, ≥98% approval rate). We sample 65 videos from the groups, evaluated by 25 unique workers, yielding 1,625 judgments. The clear improvement from (B) to (C) validates that our filtering pipeline substantially enhances motion quality. We have included the study layout and full results in the Appendix.C.2.
>
>     | **Method** | **Rating 1=Low, 5=Best** |
>     |---|---|
>     | random motion | 1.375 |
>     | Without filters | 2.434 |
>     | **Final-cleaned** | **4.133** |

---

> > ### Author Response · Authors · 2025-11-24
> > **Reply to reviewer 5Tyz (2/3)**
> >
> > **[Q1-Text Dataset quality]** :
> >
> > * **[Q1: Automated annotation user-study analysis]**:  In addition, we conducted a human evaluation of the generated annotations using an MTurk study that mirrors the setup shown above: workers were shown an input video and a text description, and rated (1-5) how much they agreed with the statement “The text accurately describes the hand motion in the input video.” We evaluated three conditions: (A) a random annotation, (B) our generated annotation, and (C) the corresponding human-written annotation. The results are shown below: random descriptions score very low (1.106), confirming that workers reliably detect incorrect text; our generated annotations achieve a high score of **4.244**, which is very close to **human-written descriptions (4.673)**. **This strong alignment demonstrates that our automated pipeline produces realistic, human-quality annotations that accurately describe the hand motion in the video**.
> >
> >     | **Method** | **Rating 1=Low, 5=high** |
> >     |---|---|
> >     | Random | 1.106 |
> >     | **Ours-annotation** | **4.244** |
> >     | **Human annotation** | **4.673** |
> >
> > * **[Automated annotation Quantitative analysis]**: In addition to the above newly conducted user-study, we would also highlight **Table 3** in the main paper. Following EgoHOD[2], we employ GPT-Scores to compare our generated annotations against human-written annotations and other baselines. As shown in Table 3 of the main paper, our method achieves substantially higher GPT-Scores, indicating that the annotations produced by our pipeline are both semantically accurate and closely aligned with human descriptions.
> >
> > We hope this addresses the reviewer’s concerns regarding the overall quality of the dataset.
> >
> > **[Q2 - Generating novel motions]** : The main focus of our work is to enable in-the-wild hand-motion modelling. Similar to prior LLM-based motion models such as MotionGPT, our method can interpret previously unseen textual descriptions of a seen action and generate the corresponding motion in many different varieties. However, generating entirely novel motion that has not been seen in training is challenging, as it is for other deep learning models. We will include additional qualitative examples in the supplemental video.
> >
> > **[Q2 - Generating novel composition]** :  We can generate sequences with compositions, for example, a person playing a guitar while writing in a notebook, among others and we will include these in the supplemental video. However, our dataset mostly comprises sequences containing a single dominant action with limited multi-action/auxiliary composition, which restricts the model’s exposure to such patterns during training. Incorporating more compositional motion examples or multi-action sequences into the dataset would directly benefit the model. We will include additional qualitative examples in the supplemental video.

---

> > > ### Author Response · Authors · 2025-11-27
> > > **Reply to reviewer 5Tyz : Supplemental video update**
> > >
> > > To address the reviewer’s question on generating novel motion and composition, we have expanded the supplemental material with new qualitative results, under the folders novel_composition and novel_text_description. We hope these additions address your concerns on this point.

---

> > > > ### Author Response · Authors · 2025-11-27
> > > > **Reply to reviewer 5Tyz :  Update MoMask T2M comparison**
> > > >
> > > > **[Comparison with MoMask]**: Thank you for your question regarding recent baselines. In the initial version of our work, we followed the HOIGPT (CVPR 2025) evaluation protocol and reported results on the corresponding set of baselines. Based on the reviewers’ comments, we additionally trained MoMask on our dataset for the T2M task. The results are shown below:
> > > >
> > > > |  | **T2M** |  |  |  |  |
> > > > |:---:|:---:|:---:|:---:|:---:|:---:|
> > > > | **Method** | **RP3 ↑** | **MMDist ↓** | **KID ↓** | **Diversity →** | **MultiModality ↑** |
> > > > | Ground Truth | 0.667 | 1.903 |  | 3.964 |  |
> > > > | MoMask | 0.628 | 2.247 | 0.516 | 3.793 | 1.819 |
> > > > | **Ours** | **0.721** | **1.765** | **0.216** | **3.865** | **1.984** |
> > > >
> > > > As shown, our method consistently outperforms MoMask across all metrics, indicating a stronger T2M performance. We hope this address your concerens on this point, encourages to increase the rating and confidence.

---

> ### Author Response · Authors · 2025-11-24
> **Reply to reviewer 5Tyz (3/3)**
>
> **[W2 - Geometric refinement stage]** :
>
> * **[W2a - Gumbel-softamax detail]**:  For motion decoding, our LLM predicts logits over the full text-motion vocabulary, and we explicitly slice out only the motion-token channels before decoding. These motion-specific logits are converted to categorical motion tokens using a Gumbel-Softmax operator, and the resulting relaxed tokens are decoded into continuous 3D trajectories using our SHIFT decoder. Training combines the standard LM loss with a continuous-space reconstruction loss, yielding the final objective L= α x LM loss + λ x Rec loss . This formulation ensures that motion tokens remain discrete during inference while still allowing geometric refinement through differentiable reconstruction. We thank the reviewer for raising this point and note that the full description (including motion-logit slicing, Gumbel-Softmax decoding, and the reconstruction formulation) is now clearly included in the **Appendix section A**.
>
> * **[W2b -  cross-entropy and reconstruction loss (α, λ) ]**:  We conducted a full α/λ sensitivity sweep (presented in the table below) and observe a consistent trend: **large λ (low α) smooths the motion but affects semantic alignment**, while **large α (low λ) sharpens token prediction but increases geometric artifacts** (higher KID). The balanced setting **α=0.5, λ=0.5 delivers the best overall performance** across M2T (RP3=0.721, KID=0.216) and T2M (RP3=0.571, Bleu4=0.181). When λ is high (α low), the model struggles to model the overall distribution, highlighting the significance of the LM loss. On the contrary, when α is too high (λ low), the model predicts sharp discrete tokens but yields poorer geometric realism.
>
>     | **LM (α)** | **Rec (λ)** | **RP3 ↑** | ** KID ↓** | ** RP3 ↑ ** | ** Bleu4 ↑ ** |
>     |---|---|---|---|---|---|
>     |  |  | 0.671 |  | 0.667 |  |
>     | **0** | **1** | **0.413** | **0.886** | **0.099** | **0.021** |
>     | 0.1 | 0.9 | 0.498 | 0.725 | 0.357 | 0.077 |
>     | 0.25 | 0.75 | 0.522 | 0.335 | 0.403 | 0.116 |
>     | **0.5** | **0.5** | **0.721** | **0.216** | **0.571** | **0.181** |
>     | 0.75 | 0.25 | 0.712 | 0.234 | 0.544 | 0.172 |
>     | 0.9 | 0.1 | 0.708 | 0.289 | 0.543 | 0.171 |
>     | **1** | **0** | **0.69** | **0.297** | **0.568** | **0.173** |
>
> * **[W2c - GR vs EgoLM blending]**: We thank the reviewer for raising this point. As shown in Table 5, the **EgoLM soft-blending setup performs much closer to our model without geometric refinement** (w/o GR) than to our full model with GR. For example, in T2M, EgoLM achieves 0.705 RP3 / 0.263 KID, which is more closer to the w/o GR baseline (0.690 RP3 / 0.297 KID) but noticeably weaker than our full PT+GR+IFT model (0.721 RP3 / 0.216 KID). A similar trend appears in M2T, where EgoLM obtains 0.570 RP3 / 0.171 Bleu4, again much closer to w/o GR (0.568 / 0.173) and clearly below our final GR-enhanced model (0.571 / 0.181).
>
>     This trend reflects a fundamental difference between soft-blending and our geometric-refinement stage: **soft-blending encourages smooth interpolation between latent codes**, whereas **GR explicitly optimizes motion tokens that increases the fidelity in continuous motion space**. Furthermore, soft-blending interacts poorly with cross-entropy, since CE encourages sharp token boundaries while blending pulls representations toward smoothed mixtures, leading to limited gains. This design is why GR consistently yields stronger results across both T2M and M2T.
>
> We hope our responses address your major concerns and help increase your rating and confidence.
>
> Reference:
>
> [1] Zhang et al. "HaWoR: World-Space Hand Motion Reconstruction from Egocentric Videos", 2025.
>
> [2] Pei et al. "Modeling Fine-Grained Hand-Object Dynamics for Egocentric Video Representation Learning", 2025.

---

### Official Review · Reviewer_kcC7 · 2025-11-01

**Soundness:** 3
**Presentation:** 3
**Contribution:** 2
**Rating:** 6
**Confidence:** 4

**Summary:**

The work introduces a large-scale dataset of human hands automatically extracted from a large-scale egocentric video dataset, consisting of motion sequences and accompanying textual activity descriptions. Additionally, a method for text-to-hand trajectory generation and hand trajectory-to-text description is proposed, benefiting from a novel hand tokenization scheme. The proposed method outperforms multiple baselines.

**Strengths:**

The scale of the hand motion dataset is unprecedented. The work provides a thorough analysis of the introduced dataset. The proposed method outperforms multiple baselines on the introduced dataset. The design choices of the method and filtering of the dataset are quantitatively supported by ablation studies. The proposed method supports both the text-to-motion and motion-to-text tasks simultaneously.

**Weaknesses:**

For a work introducing a novel dataset as its main contribution, more qualitative examples of the generated trajectories as well as hand poses and coarse/fine-grained textual descriptions are necessary. This is a major weakness, especially coupled with the following concern:
The noun distribution in Figure 11 shows several undesirable entries being common in the dataset, e.g. "hand" (hand touching a hand?) and "cut" (a verb?). This raises questions about the quality of the dataset's noun/verb annotations.
The efficacy of the method is supported by its strong performance against baselines. However, at no point does the work mention any human verification of the generated dataset. As such, it is difficult for a reviewer to ascertain its quality. A human study would have greatly benefited the work. The contribution from the dataset side is thus limited for me.

In addition to insufficient qualitative examples of the dataset, more qualitative examples of the method's output must be included.

The proposed method was only evaluated on the introduced dataset, and not on other datasets such as GigaHands or ARCTIC.

It would be good to add qualitative examples of trajectories to the rightmost t-SNE plot in Fig. 6. Merely covering a broader t-SNE range of hand poses could also be achieved by a large fraction of erroneous poses in the dataset.

The object is not at all considered in the proposed dataset and method, limiting their usefulness.

The work could benefit from a table comparing it to existing datasets in terms of scenario count, total length, diversity (number of objects), etc.

**Questions:**

In Section 3.3, what is meant by "top-200" and "top-3000"?

What is the purpose of the introduced "testing" split if numbers are reported on the validation split only?

---

> ### Author Response · Authors · 2025-11-25
> **Reply to reviewer kcC7 (1/3)**
>
> We thank you for your helpful comments and feedback. We appreciate your positive remarks that “the scale of the hand motion dataset is unprecedented” and that “the proposed method outperforms multiple baselines.” We address your remaining concerns below.
>
> We address each of your questions and concerns below.
>
> **[W1-Dataset]** :
>
> * **[Noun/Verb distribution]** : We thank the reviewer for the insightful observation and question. The noun/verb distribution in Figure 11 is computed over lemmatized text annotations. For instance, “hand” primarily comes from the fine-grained annotations, where annotations explicitly describe actions of the left or right hand (e.g., “holds the fabric with the left hand…”). Likewise, expressions such as “cutting board” or “cutting tool” are lemmatized to the noun “cut”. These are not annotation errors, rather a natural result of the annotation style and the lemmatization process.
>
> * **[ Annotation user-study analysis]**:  In addition, we conducted a human evaluation of the generated annotations using an MTurk study that mirrors the setup shown above: workers were shown an input video and a text description, and rated (1-5) how much they agreed with the statement “The text accurately describes the hand motion in the input video.” We evaluated three conditions: (A) a random annotation, (B) our generated annotation, and (C) the corresponding human-written annotation. The results are shown below: random descriptions score very low (1.106), confirming that workers reliably detect incorrect text; our generated annotations achieve a high score of **4.244**, which is very close to **human-written descriptions (4.673)**. **This strong alignment demonstrates that our automated pipeline produces realistic, human-quality annotations that accurately describe the hand motion in the video**. We have included the study **layout and full results in the Appendix.C.2**.
>
>     | **Method** | **Rating 1=Low, 5=high** |
>     |---|---|
>     | Random | 1.106 |
>     | **Ours-annotation** | **4.244** |
>     | **Human annotation** | **4.673** |
>
> * **[Motion-Reconstruction User-Study]**: Following the reviewer’s suggestion, we conducted an additional MTurk user-study to assess the perceptual quality of our reconstructed hand motions. The workers are shown the input egocentric video alongside two rendered 3D hand-motion animations (front and back views) and rated, on a 1-5 Likert scale, how realistic the 3D motion was and how closely it resembled the motion in the video. We evaluated three setups: (A) random motions sampled from unrelated sequences, (B) our reconstruction without filters, and (C) our final filtered reconstruction. As we can see from table below, **users strongly preferred our final reconstruction (4.133) compared to the unfiltered version (2.434) and the random baseline (1.375)**. We restricted participation to experienced Turkers (>5000 HITs, ≥98% approval rate). We sample 65 videos from the groups, evaluated by 25 unique workers, yielding 1,625 judgments. The clear improvement from (B) to (C) validates that our filtering pipeline substantially enhances motion quality. We have included the study layout and full results in the Appendix.C.2.
>
>     | **Method** | **Rating 1=Low, 5=Best** |
>     |---|---|
>     | random motion | 1.375 |
>     | Without filters | 2.434 |
>     | **Final-cleaned** | **4.133** |
>
> * **[top-200 and top-3000 clarification]** : Thank you for the question. In Section 3.3, we rank the samples based on increasing order of how diverse it is and use the “top-200” and “top-3000”samples for visualization to improve interpretability. Plotting all 30K ego-videos from our dataset together with 16K samples from GigaHands and 700 from GRAB produces visually congested and unreadable t-SNE maps. Therefore, for trajectory analysis, we rank samples by their diversity relative to the dataset and visualize only the top-200 most representative ones. Similarly, for hand-pose distribution we report only the top-3000 samples to maintain a clear and interpretable plot. This is similar to how text-annotation papers visualize only the top-n most frequent verbs or nouns to improve readability; our use of top-n diverse samples follows the same principle.
>
> **[W1 & W2 - Qualitative results]** :We thank the reviewer for the suggestion to include more qualitative examples. In the current supplemental video, we present 8 representative dataset examples with both coarse and fine-grained annotations (timestamp 02:45-03:00) and 14 qualitative model results (7 T2M and 7 M2T), including comparison sequences. We intentionally kept the video concise so that readers can clearly follow the key ideas without being overwhelmed. To address the reviewer’s request, we will add a separate supplemental folder containing additional qualitative videos.

---

> > ### Author Response · Authors · 2025-11-25
> > **Reply to reviewer kcC7 (2/3)**
> >
> > **[W3 - Results on public benchmarks]** :  Based on the reviewer’s comment, we perform an additional evaluation of our method on the public datasets GRAB[1] and ARCTIC[2]. We  follow the evaluation protocol of HOIGPT[5] and train all methods, including our own, on a publicly available captured dataset composed of ARCTIC and GRAB, covering 5.1K / 0.5K / 0.5K sequences for training, validation, and testing. We evaluate performance on the Text2Motion (T2M) and Motion2Text (M2T) tasks using the metrics described in Section Experiments, and report the full results in **Appendix. B.3 (Additional Experiments)**. For convenience, we provide a simplified version of the results below.
> >
> > As shown in the tables, our method consistently outperforms prior approaches across both tasks. In T2M, our model achieves the **highest R-Precision (0.492)**, the **lowest MMDist among generative models (3.008)**, and competitive KID scores, while also providing substantially better multimodality than MotionGPT and T2MGPT. Notably, HumanMDM, a diffusion-based model, tends to generate smooth but less semantically aligned motions, which is reflected in its lower R-Precision and higher MMDist in this reduced-data setting.
> > In M2T, our method again achieves the best performance across all major metrics, indicating stronger bidirectional grounding between motion and language compared to MotionGPT and TM2T. Although our model is explicitly designed for in-the-wild hand-motion modeling, it nonetheless generalizes effectively to controlled HOI datasets, demonstrating the strength and versatility of the learned representation.
> >
> > |  | **T2M** |  |  |  |  |
> > |:---:|:---:|:---:|:---:|:---:|:---:|
> > | **Method** | **RP3 ↑** | **MMDist ↓** | **KID ↓** | **Diversity →** | **MultiModality ↑** |
> > | Ground Truth | 0.525 | 2.763 |  | 4.581 |  |
> > | HumanMDM | 0.429 | 4.047 | **0.0107** | **4.915** | **2.567** |
> > | MotionGPT | 0.371 | 3.609 | 0.0409 | 3.315 | 1.955 |
> > | T2MGPT | 0.407 | 3.761 | 0.0773 | 4.956 | 1.658 |
> > | **Ours** | **0.492** | **3.008** | 0.0144 | 3.811 | 2.393 |
> >
> > |  | **M2T** |  |  |  |
> > |:---:|:---:|:---:|:---:|:---:|
> > | **Method** | **RP3 ↑** | **Bleu4 ↑** | **Bleu4 ↑** | **Rouge_L ↑** |
> > |  Ground Truth| 0.5291 |  |  |  |
> > | TM2T | 0.3519 | 0.1815 | 0.2245 | 0.5174 |
> > | MotionGPT | 0.4262 | 0.2158 | 0.5167 | 0.5278 |
> > | **Ours** | **0.4601** | **0.2341** | **0.5732** | **0.5822** |
> >
> > **[Q2 - Validation set clarification]** : We would like to clarify that all quantitative, qualitative, and user-study results are evaluated on our test set. If the reviewer can point to a specific line or table that caused this impression, we will correct the wording accordingly in the camera-ready version.
> >
> > **[W4 - TSNE with trajectory overlay]**: The reviewer raises a good point with regarding visualization of motions from different parts of the t-sne space and whether all parts of the this space represent quality meaningful motions. We thank for this suggestion. We will add visualizations of several motions that cover various locations in the t-sne space into the revised paper/ supplemental video.
> >
> > **[W6 - Table comparing it to existing datasets]** : We thank the reviewer for this valuable suggestion, we will update the comparisons to the appendix in revised version of the paper.

---

> > ### Author Response · Authors · 2025-11-26
> > **Reply to reviewer kcC7: Supplemental video update**
> >
> > To address the reviewer’s request for additional qualitative examples, we have expanded the supplemental material with new qualitative results, including: (i) dataset ground-truth sequences with coarse and fine annotations, (ii) text-to-motion generations, and (iii) motion-to-text conditioning examples. We believe these additions provide a more comprehensive qualitative understanding of the dataset and the method, and we hope they satisfactorily address your concerns on this point.

---

> ### Author Response · Authors · 2025-11-25
> **Reply to reviewer kcC7 (3/3)**
>
> **[W5 - Lack of objects]** : We thank the reviewer for their insightful comment regarding the absence of objects in our dataset. We fully agree that object-motion modelling and Hand–Object Interaction (HOI) are important directions. We believe achieving HOI in the wild will require several advances: (1) modelling in-the-wild hand motions, (2) handling arbitrary object shapes and affordances, and (3) developing a unified multi-modal framework that jointly models hands, objects, and language. CLUTCH addresses the first component by providing a strong and scalable hand-motion backbone. Even recent HOI-focused works such as HOIGPT[5] and HoiDini[6] operate only on a small, fixed set of objects and interaction scenarios from GRAB/ARCTIC. Expanding and getting more data is timing consuming and not scalable. Below, we clarify why large-scale in-the-wild HOI remains beyond current SOTA capabilities, how this aligns with limitations in prior work, and why our contribution remains meaningful.
>
> * **[Dataset HOI-Reconstruction limitations]**: Reconstructing accurate hand-object interaction in videos remains an open challenge. Existing HOI datasets such as GRAB[1] and ARCTIC[2] rely on pre-modeled or manually reconstructed objects and are therefore limited in scale. Even in controlled capture setups like GigaHands[3] with 51 cameras, hand-object interaction reconstruction remains unreliable due to effects such as occlusions, as noted by the authors in [4]. In in-the-wild settings, this difficulty increases substantially. We expect recent advances in large-scale 4D reconstruction models (e.g., SAM-3D; interaction-aware 4D Gaussians tracking) to eventually enable scalable HOI capture, but these tools are not yet mature enough to be used for building a dataset.
>
> * **[Usefulness without explicit object modeling]** Even without object geometry, our dataset and model can be employed for several real applications, including improving hand-motion priors for tracking systems, motion forecasting, animation pipelines (where artists refine hand motions independently of objects), and dense motion captioning from hand trajectories. Importantly, we also show strong transfer to ARCTIC+GRAB benchmarks **(see W3 answer)**, demonstrating robustness even in controlled HOI datasets.
>
>     In summary, HOI reconstruction remains the bottleneck, not model design and CLUTCH solves the part of the problem that is tractable today. Once large-scale HOI reconstruction tools become available, CLUTCH can be extended with object-shape tokens for full HOI-LLM integration.
>
>
> **[W1-Dataset as main contribution]** : Our goal is to address the core limitations of existing motion models and take a first step toward scalable, foundation-level 3D motion modelling. A central observation motivating our work is that, unlike text, image, or VLA foundation models—trained on hundreds of millions of images (e.g., TransFusion) or tens of thousands of hours of video (e.g., π₀), the 3D hand-motion domain lacks datasets of comparable scale or diversity. In addition, there are other open challenges like, what is the right in-the-wild hand-motion representation, how to handle the diversity of the motion data at such scale.
>
> Our work focuses on solving these foundational problems: building large-scale in-the-wild motion data using VLM+LLM tools, developing the SHIFT tokenizer to represent unconstrained hand motion effectively, and modelling these tokens within an LLM for bidirectional T2M/M2T reasoning. The resulting LLM-based motion framework achieves strong performance across realism, reconstruction, and diversity, demonstrating that such tokenized modelling is viable at scale. We believe our results will serve as a stepping-stone toward general and broadly applicable motion-foundation models.
>
> We hope this addresses your concerns fully and encourages you to raise your rating and confidence.
>
> References:
>
> [1] Taheri et al., "GRAB: A Dataset of Whole-Body Human Grasping of Objects", 2020.
>
> [2] Fan et al., "ARCTIC: A Dataset for Dexterous Bimanual Hand-Object Manipulation", 2023.
>
> [3] Fu et al., "GigaHands: A Massive Annotated Dataset of Bimanual Hand Activities", 2025.
>
> [4] https://github.com/brown-ivl/GigaHands/issues/5
>
> [5] Huang et al., "HOIGPT: Learning Long Sequence Hand-Object Interaction with Language Models", 2025.
>
> [6] Ron et al., "HOIDiNi: Human-Object Interaction through Diffusion Noise Optimization", 2025

---

> ### Author Response · Authors · 2025-11-27
> **reply to reviewer kcC7: table comparison with existing datasets**
>
> Thank you for the insightfull suggestion on adding scenario's and comparison with existing datasets.  Please find the comparison in the below presented tables, prior datasets such as ARCTIC, TACO, OakInk2, HOT3D, and GigaHands are all studio-captured and focus on a limited set of objects and activities, whereas our 3D-HIW dataset is a large-scale in-the-wild hand-motion dataset. It contains 5,000 minutes, 32.7K sequences, 9M poses, and a much broader semantic range (1,355 objects and 1,045 verbs).
>
> | Name | Setting | Markerless | Hand Track | Object Track | #mins | #motions | #poses | #views | #frames | #frame per view | #subjects | #objects | #verbs |
> |---|---|---|---|---|---|---|---|---|---|---|---|---|---|
> | ARCTIC | studio | ✗ | mocap | mocap | 121 | 339 | 218k | 9 | 2.1M | 233k | 10 | 11 | ✗ |
> | TACO | studio | ✗ | mocap | mocap | 202 | 2.3k | 363k | 13 | 4.7M | 361k | 14 | 196 | 13 |
> | OakInk2  | studio | ✗ | mocap | mocap | 557 | 2.8k | 993k | 4 | 4.01M | 1M | 9 | 75 | 55 |
> | HOT3D  | studio | ✗ | mocap | mocap | 833 | 4.1k | 1.7M | 2–3 | 3.7M | 1.8M | 19 | 33 | ✗ |
> | GigaHands | studio | ✓ | auto | auto | 2,034 | 13.9k | 3.7M | 51 | **183M** | 3.5M | **56** | 417 | **1467** |
> | Ours (3D Hands-in-the-wild) | in-the-wild | ✓ | auto | / | **5,000** | **32.7K** | **9M** | 1 | 12M | **12M** | / | **1355** | 1045 |
>
> We also include a breakdown of the top activity clusters (e.g., cutting food, knitting, repairing, cooking, painting, typing, gardening), illustrating the diversity of real-world scenarios represented in our data.
>
>
> | Cluster | Scene Label       | Raw Keywords / Pattern Examples            | Count | Example Annotation Sentence |
> |--------|--------------------|---------------------------------------------|-------|------------------------------|
> | 1      | cutting_food       | cut, chop, slice, peel                      | 6184  | The person cuts vegetables with a kitchen knife. |
> | 2      | knitting           | knit, yarn, needle, crochet, hook           | 4059  | The person knits a piece of fabric using two needles. |
> | 3      | repair             | engine, machine, tool, repair               | 3309  | The person uses a tool to adjust a part of a machine. |
> | 4      | stirring_cooking   | stir, mix, cook, pan, fry                   | 2714  | The person stirs ingredients in a frying pan. |
> | 5      | painting           | paint, brush                                | 1806  | The person paints a wooden surface with a brush. |
> | 6      | laptop_keyboard    | laptop, keyboard, type, trackpad, computer  | 1357  | The person types on a laptop keyboard. |
> | 7      | sewing             | sew, fabric, thread, stitch                 | 1286  | The person sews fabric together using a needle and thread. |
> | 8      | phone_use          | phone, screen, swipe, tap                   | 1048  | The person taps on a smartphone screen. |
> | 9      | food_prep_general  | bowl, spoon, egg, marinate, knead           | 771   | The person mixes ingredients in a bowl with a spoon. |
> | 10     | clay_crafting      | clay, mold, pottery                         | 691   | The person shapes a ball of clay with both hands. |
> | 11     | cleaning           | wash, clean, scrub                          | 657   | The person scrubs a surface with a sponge. |
> | 12     | gardening          | plant, soil, seedling                       | 327   | The person plants a small seedling into soil. |
> | 13     | vehicle_operation  | vehicle, steering                           | 216   | The person turns the steering wheel of a vehicle. |
> | 14     | pouring_liquids    | pour, water, liquid                         | 470   | The person pours water from a bottle into a cup. |
> | 15     | card_interaction   | card, shuffle, plays                        | 831   | The person shuffles a deck of cards. |
> | 16     | scissors_cut       | scissors, cut                               | 50    | The person cuts paper using scissors. |
> | 17     | drinking           | drink, glass                                | 80    | The person drinks water from a glass. |
> | 18     | music              | guitar, strum, fret, piano, keys            | 158   | The person strums chords on a guitar. |
> | 19     | crafting           | cardboard, marking, shaping                 | 339   | The person shapes a piece of cardboard with their hands. |
> | 20     | other              | adjust, examines, open, flip                | 5038  | The person adjusts and examines a small object in their hands. |
>
>
> We hope this directly addresses the reviewer’s request and highlights that 3D-HIW is significantly larger and more diverse than existing 3D hand motion datasets.

---

### Official Review · Reviewer_oQBC · 2025-11-01

**Soundness:** 3
**Presentation:** 2
**Contribution:** 3
**Rating:** 6
**Confidence:** 3

**Summary:**

The researchers present two main contributions:
1. A large-scale dataset, 3D Hands in the Wild (3D-HIW): They construct this dataset using 3D hand trackers to extract hand trajectories from egocentric videos and leverage vision-language models to generate corresponding textual annotations.
2. The CLUTCH model: A transformer-based model trained on 3D-HIW, introducing two key innovations: (a). SHIFT: A novel tokenization method that decomposes hand motions into separate trajectory and pose components for each hand, improving generalization and yielding more accurate motion reconstructions. (b). Geometric Refinement: A fine-tuning stage applied to the language model that enhances the geometric accuracy and realism of the generated animations. This includes a reconstruction loss directly applied to the decoded 3D motion parameters.

**Strengths:**

1. The motion dataset represents a highly valuable contribution to the field.
2. The SHIFT mechanism is well-motivated, effectively decoupling trajectory-level movements from fine-grained finger motions, and ablation studies demonstrate its effectiveness. Also, enabling bidirectional motion–text decoding is an innovative design, and it is noteworthy that this approach performs successfully in practice.

**Weaknesses:**

1. Since the hand motions are synthesized from a single textual description, how is motion diversity ensured? Are there mechanisms in CLUTCH to generate varied hand trajectories or poses from the same caption?
2. While the proposed approach is effective for isolated hand motions and the datasets are centered on hand-only movements, its omission of object interactions could constrain its applicability to more realistic, object-involved settings.

**Questions:**

1. What "Div →" means ?
2. The presentation could be improved, for example, inconsistent title capitalization (Line458), text size in tables, and line-breaking logic (Line271).
3. Because text is discrete while motion is continuous, it may be more natural to model hand motions using architectures like TransFusion, similar to how VLMs such as Pi0?

---

> ### Author Response · Authors · 2025-11-24
> **Reply to reviewer oQBC (1/2)**
>
> Thank you for your helpful comments and feedback. We appreciate your positive assessment that the **“dataset represents a highly valuable contribution to the field,”** that **“the SHIFT mechanism is well-motivated and effectively decouples trajectory-level movements from fine-grained finger motions.”**
>
> We address your remaining concerns below:
>
> **[W1 - Generating multiple samples]** : Similar to standard LLMs, CLUTCH can generate multiple diverse hand-motion sequences from the same text prompt by conditionally sampling from the token distribution during generation. We quantify this using the standard **Multi-Modality(MM)** metric and presented the results in **Table 1**. For clarity, we provide a simplified comparison of table below.
>
> As shown in Table below, CLUTCH achieves a strong **MM score (1.984)**, demonstrating that it can produce varied motion samples, while still maintaining high text alignment than other baselines (as reflected in our RP3, MMD, and KID results). MotionGPT reports a slightly higher MM value (difference of only 0.031), which comes at the cost of semantic consistency (low RP3).
>
> | **Method**      | *RP3 ↑*            | *MM ↑*             |
> |-----------------|--------------------|---------------------|
> | Ground Truth    | 0.667 ± 0.004      | –                   |
> | HumanMDM        | 0.694 ± 0.005      | 1.748 ± 0.069      |
> | MotionGPT       | **0.573 ± 0.009**  | **2.015 ± 0.095**   |
> | T2M-GPT         | 0.683 ± 0.005      | 1.892 ± 0.085       |
> | **Ours**        | **0.721 ± 0.004**  | **1.984 ± 0.084**   |
>
> We will also add more diverse samples and visual results to the supplemental video, and we will include these in the final version of the paper for improved clarity.
>
> **[Q1 - "Div →” ]** : refers to the Diversity metric, where the **arrow (→)** indicates that **values closer to the Ground Truth are better**. Please find a simplified version of the metric below. Similar to MotionGPT[1], we employ diversity to measure the overall variance of generated motions. It is computed by taking two random subsets of generated motions, extracting their feature vectors, and averaging the L2 distances between them. For more details, please check out Action2Motion[2]. To improve the clarity of our work, we will include a more detailed version of the metrics in the Appendix.
>
> **[Q2 - Inconsistent capitalization]** : Thank you for pointing these issues. We have fixed many of them in the revised version, and we will address the rest in the camera-ready version.
>
> **[Q3 - LLM + Diffusion ]**:  We agree that a diffusion-motion model integrated with a VLM/LLM represents a highly promising and rapidly evolving research direction. However, both continuous (diffusion-based) and discretized (tokenizer-based) representations are valid modeling choices, as demonstrated across prior works.  Recent work such as DDT-LLaMA[3] (CVPR 2025 Oral) shows that diffusion timesteps can be used not only as a generative prior but also as an effective tokenizer, producing discrete visual tokens and still achieve strong results. This suggests that discretization remains a competitive and practical strategy. In our work, we choose to tokenized motion representation, since because it has been shown to work well across different modalities—including audio (e.g., SesameAI[4]), motion (HoiGPT[5], EgoLM[6]), and images. Furthermore, it allows for natural bidirectional reasoning (T2M and M2T) within the same unified token space. Nevertheless, hybrid diffusion–LLM approaches are are indeed promising, and an interesting next step to explore.

---

> > ### Author Response · Authors · 2025-11-24
> > **Reply to reviewer oQBC (2/2)**
> >
> > **[W2 - Object interaction]** :  We thank the reviewer for their insightful comment regarding the absence of object interaction modeling in our dataset and method. We fully agree that HOI modeling is an important direction. We believe that achieving HOI modeling in the wild will require several advances: (1) Modelling in-the-wild hand motions (2) Handling arbitrary objects shape and affordances in a motion model (3) developing a unified multi-modal framework that jointly models hands, objects, and language. CLUTCH addresses the first component by providing a strong and scalable hand-motion backbone. In our work, by in-the-wild we refer to unconstrained capture conditions (including diverse backgrounds, lighting, occlusion, non-scripted human behavior).  Below, we clarify why large-scale in-the-wild HOI is beyond the scope of current SOTA methods; how this aligns with limitations in prior work;  and why our contribution still remains meaningful.
> >
> > * **[Dataset HOI-Reconstruction limitations]**: Reconstructing accurate hand-object interaction in videos remains an open challenge. Existing HOI datasets such as GRAB[7] and ARCTIC[8] rely on pre-modeled or manually reconstructed objects and are therefore limited in scale. Even in controlled capture setups like GigaHands[9] with 51 cameras, hand-object interaction reconstruction remains unreliable due to effects such as occlusions, as noted by the authors in [10]. In in-the-wild settings, this difficulty increases substantially. We expect recent advances in large-scale 4D reconstruction models (e.g., SAM-3D; interaction-aware 4D Gaussians tracking) to eventually enable scalable HOI capture, but these tools are not yet mature enough to be used for building a dataset.
> >
> > * **[Current HOI models]** Even the most recent concurrent works like, HOIGPT[5] and HoiDini[11], works on a limited set of objects and scenarios from GRAB and ARCTIC dataset. Expanding and getting more data is timing consuming and not scalable. Our works focus on expanding the hand motion part of the pipeline to more diverse unconstrained settings.
> >
> > * **[Usefulness without explicit object modeling]** Even without object geometry, our dataset and model can be employed for several real applications, including improving hand-motion priors for tracking systems, motion forecasting, animation pipelines (where artists refine hand motions independently of objects), and dense motion captioning from hand trajectories. Importantly, we also show strong transfer to ARCTIC+GRAB benchmarks (see W2 answer), demonstrating robustness even in controlled HOI datasets.
> >
> > In summary, HOI reconstruction remains the bottleneck, not model design and CLUTCH solves the part of the problem that is tractable today. Once large-scale HOI reconstruction tools become available, CLUTCH can be extended with object-shape tokens for full HOI-LLM integration.
> >
> > We hope our responses address your concerns and help increase your rating and confidence.
> >
> > References:
> >
> > [1] Jiang et al., "Motiongpt: Human motion as a foreign language", 2024.
> >
> > [2] Guo et al., "Action2motion: Conditioned generation of 3d human motions", 2020.
> >
> > [3] Pan et al., "Generative Multimodal Pretraining with Discrete Diffusion Timestep Tokens", 2025.
> >
> > [4] https://github.com/SesameAILabs/csm
> >
> > [5] Huang et al., "HOIGPT: Learning Long Sequence Hand-Object Interaction with Language Models", 2025.
> >
> > [6] Hong et al., "EgoLM: Multi-Modal Language Model of Egocentric Motions", 2025.
> >
> > [7] Taheri et al., "GRAB: A Dataset of Whole-Body Human Grasping of Objects", 2020.
> >
> > [8] Fan et al., "ARCTIC: A Dataset for Dexterous Bimanual Hand-Object Manipulation", 2023.
> >
> > [9] Fu et al., "GigaHands: A Massive Annotated Dataset of Bimanual Hand Activities", 2025.
> >
> > [10] https://github.com/brown-ivl/GigaHands/issues/5
> >
> > [11] Ron et al., "HOIDiNi: Human-Object Interaction through Diffusion Noise Optimization", 2025

---

> > > ### Author Response · Authors · 2025-11-26
> > > **Reply to reviewer oQBC: Supplemental video update**
> > >
> > > **[W1 - Generating multiple samples]**: Regarding the question on generating multiple samples from a single input prompt, we have now uploaded additional supplemental videos demonstrating multiple motion generations from the same text prompt. We hope that our clarification, together with the newly added supplemental videos, adequately addresses your concerns on this point.

---

> > > ### Comment · Reviewer_oQBC · 2025-11-27
> > > **Response to authors**
> > >
> > > Thank you for the authors' efforts in responding to my comments. The explanation regarding diversity is satisfactory. However, the lack of explicit human–object interaction modeling remains a significant limitation. since HOI is crucial for realistic behavior understanding. Given the current contribution, I will keep my weak accept.

---

> ### Author Response · Authors · 2025-11-28
> **Reply to reviewer oQBC**
>
> Thank you for your thoughtful assessment and for taking the time to review our work. We agree that full HOI modeling in the wild remains a challenging problem. As R1 also noted, robust HOI tracking remains a “great challenge” for current techniques. One important aspect of this is expanding the hand motion part of the HOI pipeline to more diverse unconstrained settings, which is the main focus of our work.
>
> We also sincerely appreciate your positive remarks that “the motion dataset represents a highly valuable contribution,” that “the SHIFT mechanism is well-motivated and supported by ablations,” and that “the bidirectional motion–text design performs successfully in practice.” Your feedback reinforces our view that addressing large-scale in-the-wild hand motion is an important step toward more complete HOI modeling.

---

### Official Review · Reviewer_x3VX · 2025-11-06

**Soundness:** 3
**Presentation:** 3
**Contribution:** 3
**Rating:** 4
**Confidence:** 4

**Summary:**

This work presents a system for text-conditioned 3D hand motion generation and captioning in in-the-wild settings. The two primary contributions:
1. A large-scale dataset of 32,000 3D hand-motion sequences paired with textual descriptions, sourced from egocentric videos (Ego4D).
2. An LLM-based system that models motion tokens. A new VQ-VAE tokenizer that decomposes hand motion into separate codebooks for trajectory and pose
The authors demonstrate their experimental results on their new benchmark for both text-to-motion and motion-to-text tasks.

**Strengths:**

1. The 3D-HIW dataset is a good contribution. The proposed VLM-based annotation pipeline is a clever and scalable approach to captioning this in-the-wild data.
2. The paper does a good job of validating its design choices with the ablations for the SHIFT tokenizer and the training stages.
3. The paper is well-written, the figures are informative, and the core ideas are articulated clearly.

**Weaknesses:**

1. Lack of Hand-Object Interaction (HOI) is the most significant limitation. The paper frames its work as "in-the-wild"  yet the model only generates 3D hand motion. It does not model the objects being interacted with. True in-the-wild motion is almost entirely defined by HOI, which is explicitly left as future work.
2. The model is trained and evaluated exclusively on the authors' new dataset. It is unclear how CLUTCH would perform on other public benchmarks.

**Questions:**

1. How much computational resource and time does it take to generate a dataset of this size? It's key to show whether the proposed method is scalable.
2. The paper says currently their current method doesn't support HOI generation. How is Figure 1 generated? Do you manually align the hand trajectory to fit the objects?

---

> ### Author Response · Authors · 2025-11-24
> **Reply to reviewer x3VX (1/2)**
>
> We thank the reviewer for their constructive and encouraging feedback. We are glad that you found the 3D-HIW dataset to be a valuable contribution and that our VLM-based annotation pipeline is a scalable solution for captioning in-the-wild data. We also appreciate the recognition of our ablation studies validating the SHIFT tokenizer and multi-stage training design.
>
> We address the remaining concerns below:
>
> **[W1 & Q2 - HOI]**:  We thank the reviewer for their insightful comment regarding the absence of hand-object interaction (HOI) modeling in our dataset and method. We fully agree that HOI modeling is an important direction. We believe that achieving HOI modeling in the wild will require several advances: (1) Modelling in-the-wild hand motions (2) Handling arbitrary objects shape and affordances in a motion model (3) developing a unified multi-modal framework that jointly models hands, objects, and language. CLUTCH addresses the first component by providing a strong and scalable hand-motion backbone. In our work, by in-the-wild we refer to unconstrained capture conditions (including diverse backgrounds, lighting, occlusion, non-scripted human behavior).  Below, we clarify why large-scale in-the-wild HOI is beyond the scope of current SOTA methods; how this aligns with limitations in prior work;  and why our contribution still remains meaningful.
>
> * **[Dataset HOI-Reconstruction limitations]**: Reconstructing accurate hand-object interaction in videos remains an open challenge. Existing HOI datasets such as GRAB[1] and ARCTIC[2] rely on pre-modeled or manually reconstructed objects and are therefore limited in scale. Even in controlled capture setups like GigaHands[3] with 51 cameras, hand-object interaction reconstruction remains unreliable due to effects such as occlusions, as noted by the authors in [4]. In in-the-wild settings, this difficulty increases substantially. We expect recent advances in large-scale 4D reconstruction models (e.g., SAM-3D; interaction-aware 4D Gaussians tracking) to eventually enable scalable HOI capture, but these tools are not yet mature enough to be used for building a dataset.
>
> * **[Current HOI models]** Even the most recent concurrent works like, HOIGPT[5] and HoiDini[6], works on a limited set of objects and scenarios from GRAB and ARCTIC dataset. Expanding and getting more data is timing consuming and not scalable. Our works focus on expanding the hand motion part of the pipeline to more diverse unconstrained settings.
>
> * **[Usefulness without explicit object modeling]** Even without object geometry, our dataset and model can be employed for several real applications, including improving hand-motion priors for tracking systems, motion forecasting, animation pipelines (where artists refine hand motions independently of objects), and dense motion captioning from hand trajectories. Importantly, we also show strong transfer to ARCTIC+GRAB benchmarks (see W2 answer), demonstrating robustness even in controlled HOI datasets.
>
> In summary, HOI reconstruction remains the bottleneck, not model design and CLUTCH solves the part of the problem that is tractable today. Once large-scale HOI reconstruction tools become available, CLUTCH can be extended with object-shape tokens for full HOI-LLM integration.
>
> **[Q1 - Computation resource for dataset ]**:  Thank you for this question. Our pipeline is designed to be highly scalable: on a single A100 GPU, the end-to-end processing time is approximately **~2 minutes per sequence** (covering motion reconstruction and both stages of annotation generation). For the full dataset of 32,000 sequences, this corresponds to ≈64,000 minutes, or ≈1,067 GPU-hours on a single A100.  Because each sequence is processed independently, the pipeline is parallel and scales linearly with available compute; for instance, using 8 GPUs reduces the time to ~133 hours, and 32 GPUs reduce it to ~33 hours. In practice, when distributed across standard compute clusters, the full dataset can be generated within **3-4 days**, demonstrating that the proposed method is computationally efficient and scalable to substantially larger datasets. For instance, the authors of GRAB and ARCTIC shared a timeline of at least a year from start to end for the dataset generation.

---

> ### Author Response · Authors · 2025-11-24
> **Reply to reviewer x3VX (2/2)**
>
> **[ W2 - Results on public benchmarks]**:
> Based on the reviewer’s comment, we perform an additional evaluation of our method on the public datasets GRAB[1] and ARCTIC[2]. We  follow the evaluation protocol of HOIGPT[5] and train all methods, including our own, on a publicly available captured dataset composed of ARCTIC and GRAB, covering 5.1K / 0.5K / 0.5K sequences for training, validation, and testing. We evaluate performance on the Text2Motion (T2M) and Motion2Text (M2T) tasks using the metrics described in Section Experiments, and report the full results in **Appendix. B.3 (Additional Experiments)**. For convenience, we provide a simplified version of the results below.
>
> As shown in the tables, our method consistently outperforms prior approaches across both tasks. In T2M, our model achieves the **highest R-Precision (0.492)**, the **lowest MMDist among generative models (3.008)**, and competitive KID scores, while also providing substantially better multimodality than MotionGPT and T2MGPT. Notably, HumanMDM, a diffusion-based model, tends to generate smooth but less semantically aligned motions, which is reflected in its lower R-Precision and higher MMDist in this reduced-data setting.
> In M2T, our method again achieves the best performance across all major metrics, indicating stronger bidirectional grounding between motion and language compared to MotionGPT and TM2T. Although our model is explicitly designed for in-the-wild hand-motion modeling, it nonetheless generalizes effectively to controlled HOI datasets, demonstrating the strength and versatility of the learned representation.
>
> |  | **T2M** |  |  |  |  |
> |:---:|:---:|:---:|:---:|:---:|:---:|
> | **Method** | **RP3 ↑** | **MMDist ↓** | **KID ↓** | **Diversity →** | **MultiModality ↑** |
> | Ground Truth | 0.525 | 2.763 |  | 4.581 |  |
> | HumanMDM | 0.429 | 4.047 | **0.0107** | **4.915** | **2.567** |
> | MotionGPT | 0.371 | 3.609 | 0.0409 | 3.315 | 1.955 |
> | T2MGPT | 0.407 | 3.761 | 0.0773 | 4.956 | 1.658 |
> | **Ours** | **0.492** | **3.008** | 0.0144 | 3.811 | 2.393 |
>
>
> |  | **M2T** |  |  |  |
> |:---:|:---:|:---:|:---:|:---:|
> | **Method** | **RP3 ↑** | **Bleu4 ↑** | **Bleu4 ↑** | **Rouge_L ↑** |
> |  Ground Truth| 0.5291 |  |  |  |
> | TM2T | 0.3519 | 0.1815 | 0.2245 | 0.5174 |
> | MotionGPT | 0.4262 | 0.2158 | 0.5167 | 0.5278 |
> | **Ours** | **0.4601** | **0.2341** | **0.5732** | **0.5822** |
>
>
> We hope this addresses your concerns fully and encourages you to raise your rating and confidence.
>
> References:
>
> [1] Taheri et al., "GRAB: A Dataset of Whole-Body Human Grasping of Objects", 2020.
>
> [2] Fan et al., "ARCTIC: A Dataset for Dexterous Bimanual Hand-Object Manipulation", 2023.
>
> [3] Fu et al., "GigaHands: A Massive Annotated Dataset of Bimanual Hand Activities", 2025.
>
> [4] https://github.com/brown-ivl/GigaHands/issues/5
>
> [5] Huang et al., "HOIGPT: Learning Long Sequence Hand-Object Interaction with Language Models", 2025.
>
> [6] Ron et al., "HOIDiNi: Human-Object Interaction through Diffusion Noise Optimization", 2025

---

> ### Comment · Reviewer_x3VX · 2025-11-24
>
> I appreciate authors' comprehensive response to address my concerns and their efforts in additional experiments. After careful review of the rebuttal and other reviews, I acknowledge the limitations of current techniques and the great challenge to jointly track HOI accurately. I believe the merits of the work outweigh the drawbacks, so I have decided to increase my rating.

---

> > ### Author Response · Authors · 2025-11-24
> > **Reply to reviewer x3VX's official comment**
> >
> > We appreciate your thoughtful assessment and your reconsideration of the rating. Thank you for taking the time to review our work so carefully and for your supportive feedback.

---

### Meta-Review · Area_Chair_5XFf · 2026-01-07

**Summary:**

The paper got mixed ratings (4, 4, 6, 6).

Reviewers value the presented large-scale dataset and the ideas behind the proposed CLUTCH model.

The reviewers’ concerns include:
 (1) questions about the diversity and quality of the curated dataset;
 (2) the lack of evaluation on other public benchmarks; and
 (3) the focus on hand motion alone, without modeling object motion or full human–object interaction (HOI).

**Reviewer Concerns:**

The authors’ responses address most of the reviewers’ concerns.

The authors provide additional experiments on public benchmarks, including GRAB and ARCTIC, demonstrating improved performance of their method.

The authors also present a user study showing that the automatic annotations are closely aligned with human annotations.

In addition, the authors explain the current lack of available technology for obtaining full HOI reconstruction from in-the-wild videos, and provide further details and answers, such as more thorough comparisons between their dataset and existing datasets.

**Reviewer Scores:**

The AC expects that the authors’ rebuttal addresses most of the reviewers’ concerns.

One reviewer who initially gave a negative rating indicates an intention to raise the final score, while reviewers with positive ratings state that they intend to maintain their original evaluations.


The concerns raised by the remaining reviewers are largely aligned with those already discussed, and the AC expects that the authors’ responses are sufficiently convincing to address them.

The AC largely agrees with the reviewers’ remaining concern that including object information in the dataset to support full HOI studies is important. However, this remains a highly challenging research direction with no clear or established solutions, as also acknowledged by the authors. While incorporating object information would further strengthen the paper, the AC believes that even without this component, the current contributions and overall quality of the paper meet the acceptance threshold.

Based on these considerations, the AC recommends acceptance of this paper.

---

### Decision · Program_Chairs · 2026-01-26

Accept (Poster)